# MAGREF: Masked Guidance for Any-Reference Video Generation with Subject Disentanglement

**Yufan Deng**[1,2], **Yuanyang Yin**[2], **Xun Guo**[2], **Yizhi Wang**[2], **Jacob Zhiyuan Fang**[2],
**Shenghai Yuan**[2], **Yiding Yang**[2], **Angtian Wang**[2], **Bo Liu**[2], **Haibin Huang**[2], **Chongyang Ma**[2]
[1]Peking University Shenzhen Graduate School    [2]ByteDance

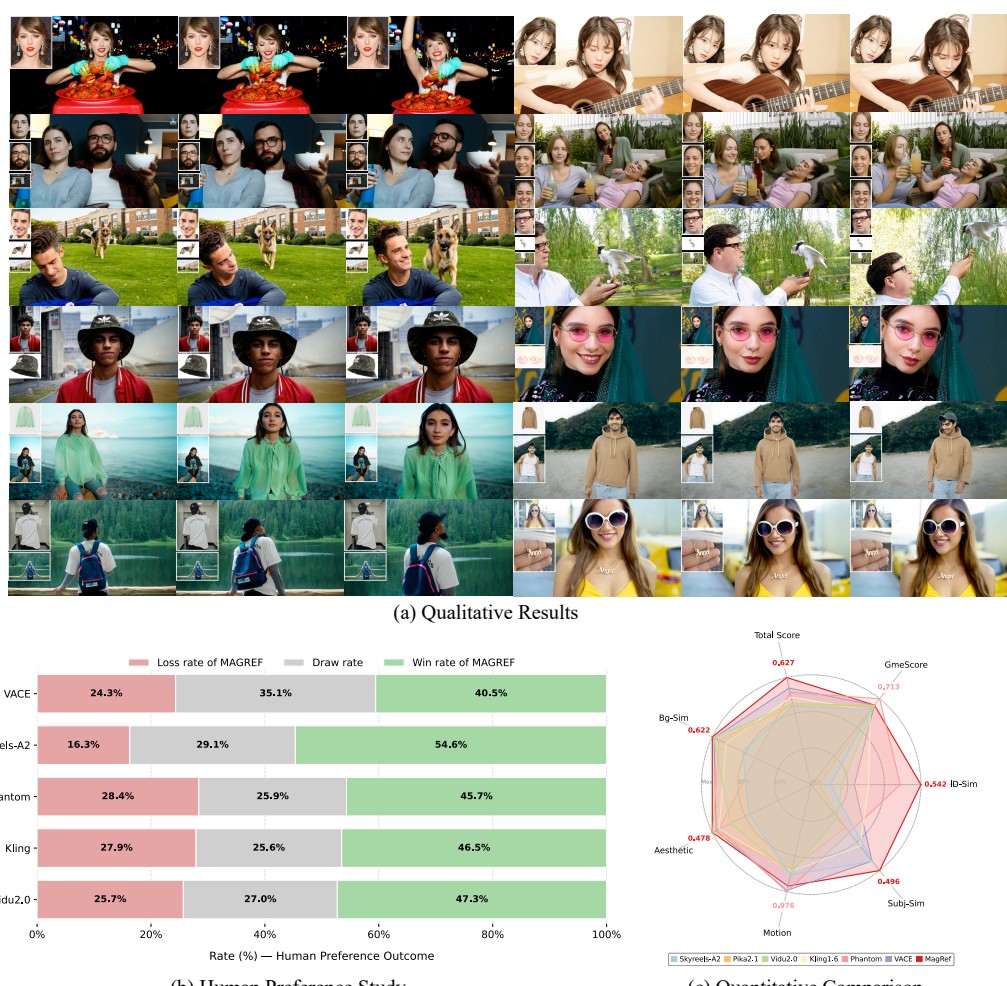

(a) Qualitative Results

(b) Human Preference Study

(c) Quantitative Comparison

Figure 1: We present **MAGREF**, a flexible video generation framework that supports arbitrary combinations of subjects including humans, animals, clothing, accessories, and environments within a single generation process, while maintaining visual consistency and faithfully following textual instructions. **(a)** Qualitative results across diverse subjects and scenes, with reference images provided in the top-left corner. More qualitative cases are provided in Figures 10–14. **(b)** User study comparing MAGREF with existing models. **(c)** Quantitative comparison for the multi-subject evaluation set.

## ABSTRACT

We tackle the task of any-reference video generation, which aims to synthesize videos conditioned on arbitrary types and combinations of reference subjects, together with textual prompts. This task faces persistent challenges, including identity

---

[1]This work was done when Yufan Deng (dengyufan10@stu.pku.edu.cn) was an intern at ByteDance.

inconsistency, entanglement among multiple reference subjects, and copy-paste artifacts. To address these issues, we introduce MAGREF, a unified and effective framework for any-reference video generation. Our approach incorporates masked guidance and a subject disentanglement mechanism, enabling flexible synthesis conditioned on diverse reference images and textual prompts. Specifically, masked guidance employs a region-aware masking mechanism combined with pixel-wise channel concatenation to preserve appearance features of multiple subjects along the channel dimension. This design preserves identity consistency and maintains the capabilities of the pre-trained backbone, without requiring any architectural changes. To mitigate subject confusion, we introduce a subject disentanglement mechanism which injects the semantic values of each subject derived from the text condition into its corresponding visual region. Additionally, we establish a four-stage data pipeline to construct diverse training pairs, effectively alleviating copy-paste artifacts. Extensive experiments on a comprehensive benchmark demonstrate that MAGREF consistently outperforms existing state-of-the-art approaches, paving the way for scalable, controllable, and high-fidelity any-reference video synthesis. The code and video demos are available in the supplementary materials.

# 1 INTRODUCTION

Recent advances in diffusion models Ho et al. (2020b); Song et al. (2020); Peebles & Xie (2023) have substantially enhanced the capability of generating realistic and temporally coherent videos, conditioned on a text prompt or a single reference image. These breakthroughs have attracted increasing attention from both academia and industry Blattmann et al. (2023a); Runway (2025); Pika (2025); OpenAI (2024), fueling a surge of interest in controllable video synthesis. Beyond text or image driven generation, there is a growing demand for leveraging multiple reference subject to provide fine-grained control over appearance and identity. This paradigm shift has sparked increasing exploration of any-reference video generation, which aims to integrate diverse visual cues into coherent, personalized, and high-fidelity video sequences.

However, conditioning video generation on both textual descriptions and multiple reference images greatly enlarges the condition space, leading to intricate interactions among arbitrary combinations of subjects, such as humans, animals, clothing, objects, and environments. This complexity makes it difficult to reliably preserve subject identities across frames, to disentangle multiple subjects without confusion, and to avoid copy-paste artifacts when integrating diverse visual cues. In particular, these complexities can be distilled into the following major challenges: **(1) identity inconsistency**, where appearance details such as facial structure or accessories fail to remain coherent; **(2) entanglement across multiple reference subjects**, where identities from different reference images are mistakenly blended or confused; and **(3) copy-paste artifacts**, which degrade visual realism and scene integrity. Recent works Yuan et al. (2024); Zhang et al. (2025); Wei et al. (2025) have shown progress in preserving a single identity, but they often rely on external identity modules and single-image references, limiting scalability in real-world applications. Other approaches Zhong et al. (2025); Jiang et al. (2025); Liu et al. (2025) simplify the conditioning process by concatenating visual tokens along the token dimension, yet these text-to-video frameworks require large-scale datasets and struggle with identity preservation and generalization. Fei et al. (2025) explores an alternative image-to-video design by concatenating references along the channel dimension with temporal masks, but still falls short in addressing the above challenges in a unified and effective manner.

To overcome these limitations, we propose MAGREF (Masked Guidance for Any-Reference Video Generation with Subject Disentanglement), which tackles them in three parts. (1) **Masked guidance for consistent multi-subject identity preservation**. we condition the model on references at the pixel level via a pixel-wise channel concatenation that preserves fine-grained appearance details, and a region-aware masking mechanism that composes a reference canvas with spatial support for each subject, enabling precise conditioning across arbitrary subject categories (humans, animals, clothing, objects, environments) within a unified architecture without structural changes. (2) **Subject disentanglement to mitigate cross-subject confusion**. We introduce a subject disentanglement mechansim that explicitly injects semantic values of subject tokens into their corresponding visual regions, thereby enforcing identity separation and reducing cross-reference confusion in any-reference video generation. (3) **A systematic four-stage data pipeline to alleviate copy-paste artifacts**. We

design an efficient data pipeline that integrates general filtering and caption, object processing, face processing, and cross-pair construction into a unified system, yielding diverse training pairs while suppressing copy-paste artifacts. Together, these components facilitate scalable, controllable, and high-fidelity any-reference video synthesis, enabling the creation of highly realistic videos.

Overall, the key contributions of MAGREF are as follows:

- We propose a unified masked guidance design that leverages a *region-aware masking mechanism* and a *pixel-wise channel concatenation* to inject references at the channel level. This preserves fine-grained appearance cues and enables precise subject conditioning across arbitrary categories, with minimal architectural modifications.

- We develop a *subject disentanglement mechanism* that injects the semantic values from text condition into their corresponding visual regions, enforcing clear separation among identities and mitigating cross-reference confusion without additional identity extraction modules.

- We establish a systematic four-stage data pipeline that constructs diverse and cross training pairs, effectively suppressing copy-paste artifacts and improving robustness. Extensive empirical evaluations show that MAGREF delivers high-quality, multi-subject consistent video synthesis, surpassing all existing approaches and achieving state-of-the-art results across several benchmarks.

## 2 RELATED WORK

**Video generation models.** Recent advancements in video generation often rely on Variational Autoencoders (VAEs) Li et al. (2024); Kingma (2013); Van Den Oord et al. (2017) to compress raw video data into a low-dimensional latent space. Within this compressed latent space, large-scale generative pre-training is conducted using either diffusion-based methods Ho et al. (2020a); Song et al. (2021) or auto-regressive approaches Yu et al. (2023a); Chen et al. (2020); Ren et al. (2025). Leveraging the scalability of Transformer models Vaswani (2017); Peebles & Xie (2023), these methods have demonstrated steady performance improvements Brooks et al. (2024); Yang et al. (2024); Blattmann et al. (2023b). This advancement significantly expands the possibilities for content generation and inspires follow-up research on text-to-video Guo et al. (2024b); Ronneberger et al. (2015); Yin et al. (2024); Yang et al. (2024); Kong et al. (2024); Wan et al. (2025); Yu et al. (2023b) and image-to-video Chen et al. (2023a); Guo et al. (2024a); Ye et al. (2023); Chen et al. (2023b); Zeng et al. (2024); Xing et al. (2024); Zhang et al. (2023); Blattmann et al. (2023b) generation models.

**Subject-driven visual generation.** Generating identity-consistent images and videos from reference inputs requires accurately capturing subject-specific features. Existing methods can be broadly divided into tuning-based and training-free approaches. Tuning-based solutions Zhou et al. (2024a); Wei et al. (2024); Chen et al. (2025); Wu et al. (2024) typically rely on efficient fine-tuning strategies, such as LoRA Hu et al. (2022) or DreamBooth Ruiz et al. (2023), to embed identity information into pre-trained models, but they require re-tuning for each new identity, limiting scalability. In contrast, training-free approaches Zhou et al. (2024b); Wang et al. (2024b) adopt feed-forward inference without per-identity fine-tuning, often enhancing cross-attention or self-attention to better preserve identity consistency.

Recent works have explored various strategies for subject-driven video generation. Some methods focus on identity preservation, such as ConsisID Yuan et al. (2024), which maintains facial consistency via frequency decomposition. Others, like ConceptMaster Huang et al. (2025) and VideoAlchemy Chen et al. (2025), leverage CLIP Radford et al. (2021) encoders together with Q-Former Li et al. (2023) to fuse visual-text embeddings for multi-concept customization. Another line of work Deng et al. (2025); Hu et al. (2025) introduces Multimodal Large Language Models (MLLMs), e.g., Qwen2-VL Wang et al. (2024a) and LLaVA Liu et al. (2023), to enhance prompt–reference interactions. Building on Wan2.1 Wan et al. (2025), methods such as ConcatID Zhong et al. (2025), VACE Jiang et al. (2025), Phantom Liu et al. (2025), and SkyReels-A2 Fei et al. (2025) further explore reference conditioning, either by concatenating image latents with noisy latents or injecting reference features as conditional inputs to guide the diffusion process.

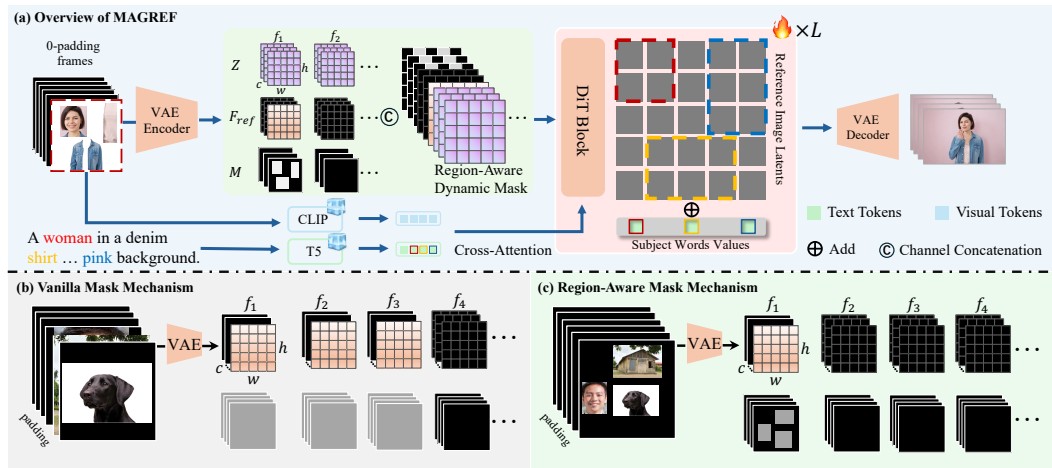

Figure 2: **(a) Overview of MAGREF.** We introduce a region-aware masking mechanism to encode multiple references and concatenate them with noise latents. *subject disentanglement* that links each reference to its textual label to avoid cross-subject entanglement. Compared with **(b) Vanilla masking mechanism**, which concatenates references along the frame dimension, our **(c) Region-aware masking mechanism** merges references into a composite image, encodes it with a VAE, and applies a downsampled binary mask to indicate subject regions, thereby better preserving first-frame consistency in I2V models.

# 3 METHOD

Given a set of reference images and a corresponding text prompt, our objective is to generate videos that preserve the consistent appearance of the specified subjects. The preliminary background on video diffusion models is provided in Appendix A. We then present our masked guidance and subject disentanglement mechanism, followed by a detailed explanation of the four-stage data curation pipeline, which decomposes video–text data and constructs diverse reference pairs.

## 3.1 VIDEO GENERATION VIA MASKED GUIDANCE

We propose MAGREF, a novel framework for coherent any-reference video generation from diverse reference images (see Figure 2). Unlike single-subject scenarios, the any-reference setting requires the model to automatically identify and align subjects with unknown number and distribution. To tackle this challenge, masked guidance mechanism introduces a region-aware masking mechanism combined with pixel-wise channel concatenation, which injects information from multiple reference images. This design enables the model to better leverage the preservation capability of the pretrained video backbone and effectively extend it to the any-reference setting.

**Region-aware masking mechanism.** To accurately incorporate multi-subject information while remaining consistent with the I2V modeling paradigm, we introduce a region-aware masking mechanism that concatenates images and simultaneously generates the corresponding region masks. Specifically, given a set of $N$ reference images $\{I_k\}_{k=1}^N$, all images are first placed onto a blank canvas at distinct spatial locations $\{p_k = (x_k, y_k)\}_{k=1}^N$. This creates a composite image $I_{\text{comp}}$, where each pixel's value is determined by the source image occupying its location. This process is formulated as:

$$I_{\text{comp}}(i, j) = \sum_{k=1}^N I_k(i - y_k, j - x_k) \cdot \mathbb{1}_{(i,j) \in R_k},$$

(1)

where $R_k$ is the rectangular region occupied by image $I_k$ on the canvas, and $\mathbb{1}_{(\cdot)}$ is the indicator function. The composite image $I_{\text{comp}}$ is treated as a single reference frame, allowing the model to inherit the native image-to-video generation capability.

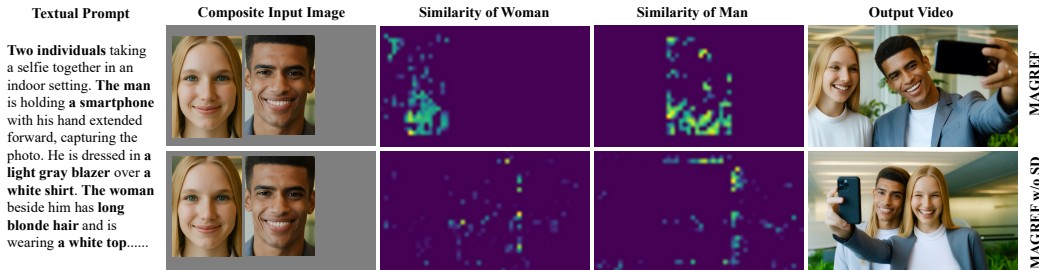

Figure 3: Cosine similarity visualization between composite reference input image and textual labels. MAGREF achieves more accurate alignment of the **Man** and the **Woman** in the multi-subject composite image with the corresponding text prompts. In contrast, removing Subject Disentanglement (SD) results in entangled and ambiguous associations.

In parallel, a binary mask is constructed to explicitly indicate the spatial regions corresponding to each subject:

$$M(i,j) = \mathbb{1}_{(i,j)\in\bigcup_{k=1}^{K} R_k}. \tag{2}$$

This mask provides a precise spatial prior of each subject in the reference frame, guiding the model to enforce strong subject-level consistency. To further improve robustness, we randomly shuffle subject locations during training to mitigate potential positional bias.

**Pixel-wise channel concatenation.** Achieving coherent and identity-consistent any-reference video generation requires precise identity-aware cues for each subject. Prior methods either inject VAE representations of reference images along the temporal dimension Jiang et al. (2025) or concatenate visual tokens after patchification Zhong et al. (2025). However, these approaches require the model to relearn identity consistency from scratch, particularly when handling varying numbers of references, which in turn demands large amounts of diverse domain data and ultimately limits generalization, leading to inconsistencies with the input images in any-reference settings. In MAGREF, rather than concatenating references along the token dimension and relying solely on self-attention Hu et al. (2025); Liu et al. (2025), we introduce a region-aware masking mechanism with pixel-wise channel concatenation, which preserves subject-specific appearance features and ensures faithful identity consistency.

Specifically, $I_{\text{comp}} \in \mathbb{R}^{1\times C_{in}\times H\times W}$ is first padded with zeros along the temporal axis to match the dimensionality of video frames, resulting in $\tilde{I}_{\text{comp}} \in \mathbb{R}^{T\times C_{in}\times H\times W}$. The padded composite is then processed by the VAE encoder $E(\cdot)$ to obtain the latent feature map:

$$F_{\text{comp}} = E(\tilde{I}_{\text{comp}}) \in \mathbb{R}^{T\times C\times H\times W}, \tag{3}$$

where $T$, $C$, $H$, and $W$ denote the number of frames, channels, height, and width, respectively. Meanwhile, the binary mask $M$ is downsampled to match the spatial resolution of $F_{\text{comp}}$ and replicated along the channel dimension, yielding $M_{\text{region}} \in \mathbb{R}^{T\times C_m\times H\times W}$. This ensures that the reference image representation is temporally aligned with the video frames, facilitating seamless integration of reference features across the entire video sequence. The raw video frames are then processed through the same VAE encoder $E(\cdot)$, producing a latent representation. Gaussian noise is added to these latents, resulting in $Z \in \mathbb{R}^{T\times C\times H\times W}$.

We concatenate the noised video latents $Z$, the reference image representation $F_{\text{comp}}$, and the feature masks $M_{\text{region}}$ along the channel dimension to construct the final input $F_{\text{input}}$:

$$F_{\text{input}} = \text{Concat}\big(Z,\ F_{\text{comp}},\ M_{\text{region}}\big) \in \mathbb{R}^{T\times(2C+C_m)\times H\times W}, \tag{4}$$

where $\text{Concat}$ denotes channel-wise concatenation. The resulting composite input $F_{\text{input}}$ is then fed into the subsequent modules of the framework to enable coherent and identity-preserving any-reference video generation.

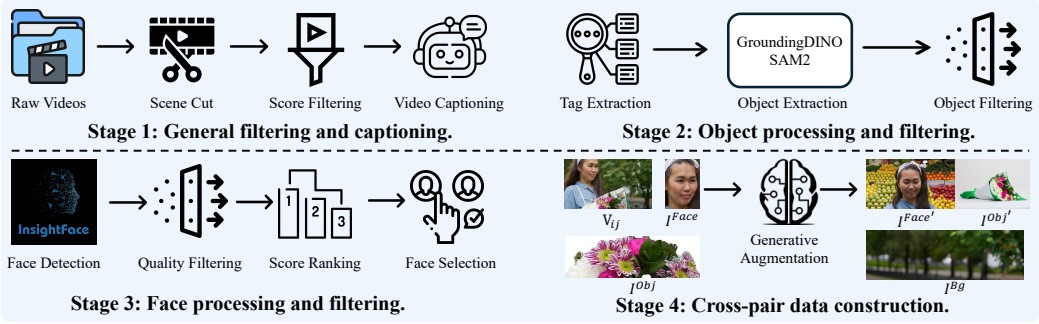

Figure 4: A systematic four-stage data pipeline for collecting high-quality training samples.

## 3.2 SUBJECT DISENTANGLEMENT

While masked guidance provides explicit feature regions for each subject and facilitates clear visual separation, aligning multiple subjects with their corresponding textual descriptions remains highly challenging. Unlike single-ID preservation, multi-subject generation requires much stronger coupling between reference images and textual conditions; otherwise, interference and entanglement across subjects are likely to occur. To address this issue, we extend the region-aware masking mechanism by explicitly associating each reference subject with its corresponding textual information.

Specifically, Subject Disentanglement begins by parsing the text condition to extract a set of word labels that correspond to the reference subject, denoted as $\{w_i\}_{i=1}^k$. For each word, we get its corresponding value embeddings $V = \{v_i\}_{i=1}^K$, $v_i \in \mathbb{R}^D$ $(i = 1, \ldots, K)$ from the cross-attention layers. To spatially anchor these semantic concepts in the visual domain, we construct a mask for each subject $M_{\text{sub}} = \{M_{sub}^k\}_{k=1}^K$ to guide the injection of the corresponding reference image value embeddings into their designated regions, where $M_{sub}^k$ is defined as:

$$M_{sub}^k(i,j) = \mathbb{1}_{(i,j) \in R_k} \in \{0,1\}^{H \times W} \quad k = 1, \ldots, K. \tag{5}$$

The subject-specific information is then directly injected into the latent representation of the first frame $z_0 \in \mathbb{R}^{1 \times C \times H \times W}$ in each layer and updated as

$$z_0' = z_0 + \alpha \sum_{i=1}^{K} \left( M_{sub}^i \odot v_i \right), \tag{6}$$

where $\odot$ denotes Hadamard (element-wise) product with broadcasting to align tensor shapes. This targeted injection operation establishes a tight alignment between the designated image regions and the associated text tokens from the very beginning of the diffusion process. As a result, it effectively mitigates attribute leakage and prevents interference across different subjects during video generation (see Figure 3).

## 3.3 FOUR-STAGE DATA CURATION

We design a systematic data curation pipeline that processes training videos, generates textual labels, and extracts reference entities including faces, objects, and backgrounds, tailored for the any-reference video generation task. As illustrated in Figure 4, the pipeline comprises four stages that progressively filter, annotate, and construct references for subsequent model training.

In Stage 1, raw videos are segmented using scene-change detection, and clips with low quality or minimal motion are discarded. The remaining clips are captioned using Qwen2.5-VL Bai et al. (2025), with a focus on motion-related content. In Stage 2, objects are identified from captions, localized with GroundingDINO Liu et al. (2024), and segmented into clean reference images using SAM2 Ravi et al. (2024). Stage 3 involves face detection with InsightFace[1], where faces are assigned identities, filtered by pose, and ranked based on quality. A fixed number of high-quality faces are selected to form the

---

[1]https://github.com/deepinsight/insightface

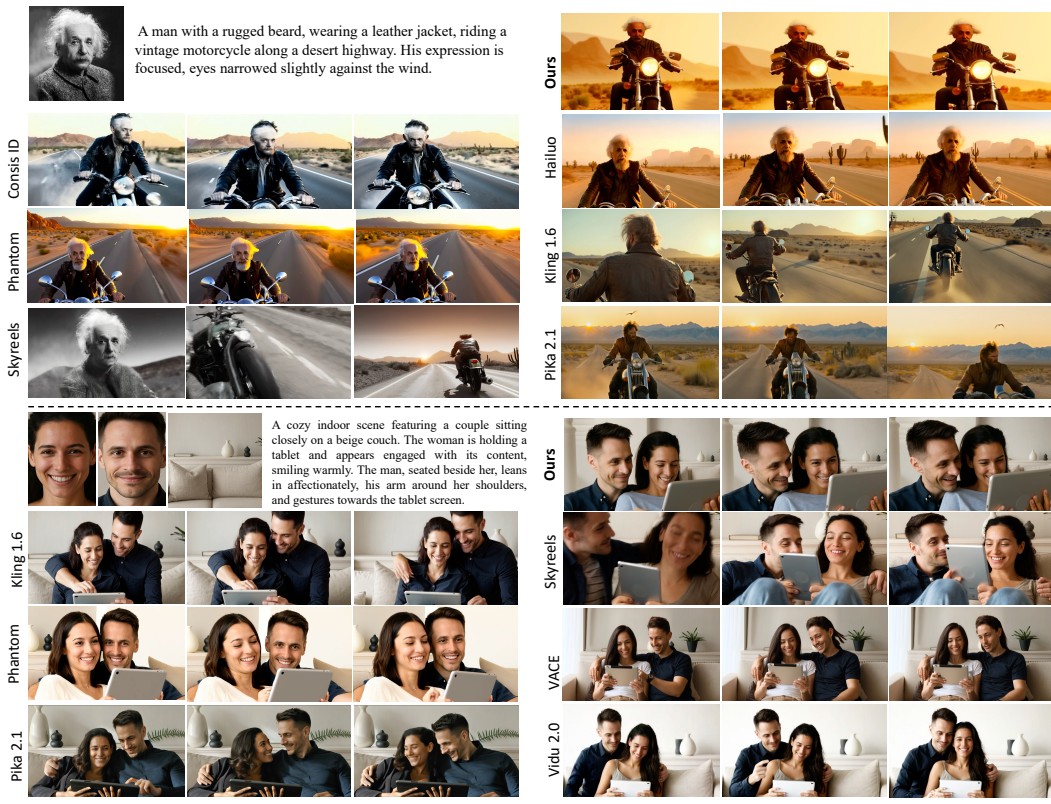

Figure 5: Comparison of our model with state-of-the-art models on single-ID (top) and multi-subject (bottom) generation. MAGREF demonstrates superior performance over other models.

reference set. Stage 4 leverages an state-of-the-art image generation model to generate augmented versions of both face and object references, introducing variations in pose, appearance, and context. Background images are also augmented to enrich the reference set. The final training sample, after all four stages, is formally defined as:

$$\mathcal{R}_i = \{\, V_i,\ C_i,\ (I_i^{\text{Face}},\ I_i^{\text{Face}'}), (I_{i,1}^{\text{Obj}},\ I_{i,1}^{\text{Obj}'}),\ \dots,\ (I_{i,k}^{\text{Obj}},\ I_{i,k}^{\text{Obj}'}),\ I_i^{\text{Bg}}\}, \tag{7}$$

where $V_i$ denotes the video clip, $C_i$ is the caption, $(I_i^{\text{Face}}, I_i^{\text{Face}'})$ are the original and transformed face references, $(I_{i,j}^{\text{Obj}}, I_{i,j}^{\text{Obj}'})$ represent the object–variant pairs, and $I_i^{\text{Bg}}$ denotes the background reference. More details of the pipeline are provided in the Appendix B.

## 4 EXPERIMENTS

### 4.1 EXPERIMENTAL SETUP

**Evaluation settings.** For the evaluation benchmark, we select a subset from prior benchmarks Yuan et al. (2024); Fei et al. (2025); Yuan et al. (2025), with the remaining cases curated to ensure diversity in subjects and scenarios. Finally, we construct a set of 120 reference-text pairs, evenly split between single-ID and multi-subject settings. Single-ID tests use one reference face image, while multi-subject tests cover diverse scenarios, including flexible combinations of two-human, three-human, and human-object-background compositions. Each case includes no more than three reference images and a natural language prompt. Detailed information is provided in the Appendix C.1.

For evaluation metrics, we consider both single-ID and multi-subject settings to comprehensively assess model performance. For single-ID evaluation, we use four metrics: (1) **ID-Sim**, cosine similarity between face embeddings, evaluating identity consistency Deng et al. (2019); (2) **Aesthetic Score**, reflecting human perceptual preferences via a predictor trained on high-quality images christoph-schuhmann (2024); (3) **Motion Smoothness**, measuring temporal coherence and motion quality Wu et al. (2023); (4) **GmeScore**, a retrieval-based vision–language alignment metric for semantic con-

Table 1: Quantitative comparison on single-ID evaluation. Best in **bold**, second best underlined.

| Model | Venue | ID-Sim | Aesthetic | Motion | GmeScore | Total Score |
|---|---|---|---|---|---|---|
| ConsisID Yuan et al. (2024) | | 0.406 | 0.418 | 0.798 | 0.720 | 0.586 |
| EchoVideo Wei et al. (2025) | | 0.455 | 0.399 | 0.782 | 0.684 | 0.580 |
| FantasyID Zhang et al. (2025) | | 0.304 | 0.456 | 0.854 | 0.726 | 0.585 |
| Concat-ID Zhong et al. (2025) | Open-source | 0.417 | 0.441 | 0.820 | 0.737 | 0.604 |
| HunyuanCustom Hu et al. (2025) | | 0.592 | 0.497 | 0.848 | 0.697 | 0.659 |
| SkyReels-A2 Fei et al. (2025) | | 0.511 | 0.443 | 0.842 | 0.618 | 0.604 |
| Phantom Liu et al. (2025) | | 0.492 | 0.504 | 0.952 | 0.722 | 0.668 |
| VACE Jiang et al. (2025) | | 0.577 | 0.524 | 0.949 | 0.696 | 0.687 |
| Hailuo Hailuo (2025) | | 0.537 | **0.527** | 0.941 | 0.714 | 0.680 |
| Pika 2.1 Pika (2025) | Proprietary | 0.301 | 0.519 | 0.851 | **0.738** | 0.602 |
| Vidu 2.0 Vidu (2025) | | 0.340 | 0.476 | 0.919 | 0.677 | 0.603 |
| Kling 1.6 Kling (2025) | | 0.359 | 0.516 | 0.846 | 0.672 | 0.598 |
| MAGREF | Ours | **0.595** | 0.516 | **0.956** | 0.710 | **0.694** |

Table 2: Quantitative comparison on multi-subject evaluation.

| Model | Venue | ID-Sim | Subj-Sim | Bg-Sim | Aesthetic | Motion | GmeScore | Total Score |
|---|---|---|---|---|---|---|---|---|
| Skyreels-A2 | | 0.274 | 0.464 | 0.507 | 0.371 | 0.884 | 0.659 | 0.527 |
| Phantom | Open-source | 0.481 | 0.364 | 0.460 | 0.458 | **0.976** | **0.713** | 0.575 |
| VACE | | 0.345 | 0.463 | 0.615 | 0.467 | 0.968 | 0.680 | 0.590 |
| Pika2.1 | | 0.239 | 0.347 | 0.596 | 0.477 | 0.851 | 0.676 | 0.531 |
| Vidu2.0 | Proprietary | 0.308 | 0.312 | 0.617 | 0.425 | 0.876 | 0.680 | 0.536 |
| Kling1.6 | | 0.387 | 0.411 | 0.571 | 0.458 | 0.864 | 0.655 | 0.558 |
| MAGREF | Ours | **0.542** | **0.496** | **0.622** | **0.478** | 0.945 | 0.681 | **0.627** |

sistency Zhang et al. (2024). For multi-subject evaluation, we introduce two additional metrics: (5) **Subj-Sim**, assessing consistency across subjects using regions extracted with GroundingDINO Liu et al. (2024) and SAM2 Ravi et al. (2024); (6) **Bg-Sim**, evaluating background consistency by an inpainting model Podell et al. (2023). Finally, we average all metrics to obtain the *Total Score*. Complete details are provided in the Appendix C.3.

**Training details.** We train our model using the FusedAdam optimizer, configured with $\beta_1 = 0.9$, $\beta_2 = 0.999$, and a weight decay of $0.01$. The learning rate is initialized at $1 \times 10^{-5}$ and follows a cosine annealing schedule with periodic restarts. To stabilize training and prevent exploding gradients, we apply gradient clipping with a maximum norm of 1.0, which benefits the optimization process. All experiments are conducted using NVIDIA H100 80GB GPUs and PyTorch. The training loss follows the standard diffusion loss formulation, as outlined in Wan et al. (2025).

## 4.2 MAIN RESULTS

**Qualitative results.** Figure 1(a) presents representative examples generated by MAGREF, with additional qualitative cases and applications provided in the Figures 10–14 of Appendix E.1. MAGREF demonstrates the ability to support arbitrary combinations of subjects including humans, animals, clothing, accessories, and environments within a single generation process, while maintaining strong consistency and faithful alignment with textual instructions.

We further compare MAGREF with state-of-the-art methods in Figure 5. MAGREF demonstrates superior identity preservation, stronger adherence to textual instructions, and better generalization in out-of-domain scenarios in both single-ID and multi-subject settings, compared to both open-source and commercial models, thus providing strong evidence of our approach's efficacy in addressing the challenges of any-reference video generation.

**Quantitative results.** We conduct a systematic evaluation of MAGREF against both open-source and proprietary models (for details on the evaluation models, see the Appendix C.2). Since some existing methods only support single-ID inputs, we report single-ID results in Table 1 and multi-

Table 3: Ablation on training paradigm and masking strategies.

| Method | ID-Sim | Subj-Sim | Bg-Sim | Aesthetic | Motion | GmeScore | Total Score |
|---|---|---|---|---|---|---|---|
| Training from T2V backbone | 0.428 | 0.403 | 0.468 | _0.450_ | _0.891_ | _0.657_ | 0.550 |
| I2V with Vanilla Masking | _0.458_ | _0.431_ | _0.492_ | 0.437 | 0.876 | 0.653 | _0.558_ |
| I2V with Regional-aware Mask (Ours) | **0.504** | **0.452** | **0.526** | **0.452** | **0.906** | **0.679** | **0.587** |

Table 4: Ablation of the entire MAGREF pipeline.

| Method | ID-Sim | Subj-Sim | Bg-Sim | Aesthetic | Motion | GmeScore | Total Score |
|---|---|---|---|---|---|---|---|
| w/o region-aware masking mechanism | 0.470 | 0.452 | 0.530 | 0.443 | 0.872 | 0.652 | 0.570 |
| w/o cross-pair data process strategy | 0.462 | 0.447 | 0.524 | 0.464 | 0.892 | 0.656 | 0.574 |
| w/o subject disentanglement mechanism | 0.493 | 0.417 | 0.518 | 0.452 | 0.919 | 0.679 | 0.580 |
| **Ours** | **0.542** | **0.496** | **0.622** | **0.478** | **0.945** | **0.681** | **0.627** |

subject results in Table 2. Across both settings, MAGREF consistently achieves the best performance in subject consistency (ID-Sim and Subj-Sim) and ranks highest in terms of overall *Total Score*. These results highlight the effectiveness of MAGREF in preserving subject identity and maintaining visual quality, while also demonstrating superior robustness across diverse evaluation scenarios.

## 4.3 ABLATION STUDIES

**Region-aware masking mechanism.** Table 3 shows that training from a T2V backbone or using vanilla masking from an I2V backbone results in reduced identity and subject consistency. In contrast, introducing the region-aware masking mechanism significantly improves performance and achieves the highest overall score. We validate all methods on a small-scale dataset with equal training steps and use the same training resources to ensure fairness. The ablation of region-aware masking mechanism on the overall pipeline in Table 4 further confirm this finding. In addition, We provide qualitative comparisons of different masking and concatenation mechanisms in Appendix D.1 and Figure 6.

**Cross-pair data processing strategy.** As reported in Table 4, removing the cross-pair data processing strategy leads to a noticeable decline in overall performance, with a particularly significant reduction in the model's ability to minimize copy-paste artifacts. This observation underscores the importance of the cross-pair augmentation strategy in addressing such artifacts. This enhanced generalization not only helps in mitigating the occurrence of copy-paste artifacts but also contributes to a more robust video synthesis process. The inclusion of cross-pair data processing improves the model's ability to handle diverse and complex scenarios, resulting in more coherent and visually consistent outputs. Overall, this strategy plays a crucial role in improving the synthesis quality, ensuring that the generated videos are more realistic and aligned with the given textual descriptions.

**Subject disentanglement mechanism.** Table 4 shows that removing the subject disentanglement mechanism leads to a noticeable decrease in both ID-Sim and Subj-Sim, weakening subject consistency. This confirms that explicitly binding each subject to its corresponding textual condition effectively reduces interference, improving multi-subject video generation quality. Additionally, Figure 3 visualizes the cosine similarity between composed reference images and their corresponding textual labels. The results show that MAGREF with the subject disentanglement mechanism aligns the Man and Woman precisely, while removing it causes entangled associations, emphasizing the importance of disentanglement for clarity in multi-subject generation. More qualitative cases analyzing the subject disentanglement mechanism can be found in the Appendix D.3.

## 4.4 USER STUDY

We conduct a user study utilizing a pairwise voting strategy to assess the performance of different video generation models. Each questionnaire consists of 60 questions, where participants are asked to compare two videos and determine which one is superior or if the two are equally good. The videos presented in each pair are randomly sampled from different models, ensuring a diverse range of comparisons. For each question, participants evaluate the videos based on three key criteria:

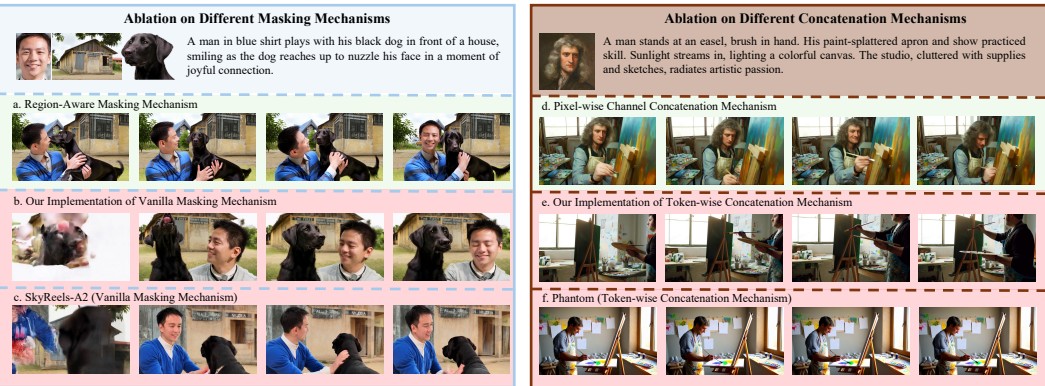

Figure 6: **Ablation study on masking and concatenation schemes**. **Left: Comparison of different masking mechanisms**. Our proposed masking mechanism maintains identity consistency and visual coherence under varying reference conditions (Top row). In contrast, the vanilla masking mechanism, which concatenates reference images along the channel dimension, results in temporal inconsistency and identity drift (The second row: our re-implementation; Bottom row: SkyReels-A2 Fei et al. (2025)). **Right: Comparison of different concatenation mechanisms**. Pixel-wise channel concatenation preserves fine-grained reference features, improving consistency with reference images. In contrast, token-wise concatenation dilutes identity-specific cues and weakens identity preservation (The second row: our re-implementation; Bottom row: Phantom Liu et al. (2025)).

identity preservation, visual quality, and alignment with the provided text description. To guarantee a comprehensive assessment, we recruit 30 experienced participants, ensuring a wide spectrum of subjective perspectives. This approach allows us to capture nuanced opinions and provide a robust evaluation. As illustrated in Figure 1(b), the results demonstrate that our method significantly outperforms existing state-of-the-art models, thereby validating the effectiveness of our approach in addressing the challenges of the any-reference video generation task.

## 5 CONCLUSION

In this work, we introduce MAGREF, a unified framework for any-reference video generation that combines pixel-wise channel concatenation with a region-aware masking mechanism, enabling the synthesis of coherent videos with multiple distinct subjects. MAGREF also incorporates a subject disentanglement mechanism and a four-stage data pipeline to enhance performance and reduce common artifacts. Extensive experiments show that MAGREF outperforms state-of-the-art methods, excelling in any-reference scenarios with strong temporal consistency and identity preservation. Future work will extend MAGREF to support unified understanding and generation using multi-modal large language models, enabling synchronized synthesis of video, audio, and text.

**Limitations and future work.** While MAGREF delivers promising perormance in any-reference video generation, it has key limitations to address in future work: it only supports image and text inputs, without integrating audio, motion or 3D spatial modalities to expand its application scope; it has not fully explored the impact of reference image quantity on performance, with limited references restricting its capture of subject variations and contextual details; its non-MLLM-based T5 text encoder struggles to capture complex text semantics, undermining text-visual alignment and video fidelity; and it lacks support for controllable long video generation, as its short-clip-optimized framework cannot maintain temporal consistency, subject identity and content coherence over extended durations.

In summary, future work will address these limitations by extending MAGREF to support multi-modal generation using advanced MLLMs, enabling synchronized synthesis of video, audio, and text, as well as the generation of long videos with controlled subject consistency and motion dynamics. By incorporating additional input modalities, optimizing the use of reference images, and improving textual understanding, we aim to further enhance the flexibility, scalability, and realism of the video generation process.

ETHICS STATEMENT

Our work adheres to the ICLR Code of Ethics. The proposed framework does not involve human or animal subjects, nor does it raise privacy, security, or legal compliance concerns. We follow proper licensing and usage guidelines. We are not aware of any potential harmful societal impacts, conflicts of interest, or bias issues introduced by this work. This research is conducted in line with standard practices for research integrity and reproducibility.

REPRODUCIBILITY STATEMENT

We are committed to ensuring the reproducibility of our results. To this end, we will release the full codebase, pretrained models, and detailed instructions for running experiments upon publication. The dataset processing pipeline and evaluation metrics are described in detail in the main paper and appendix, and additional implementation details are included in the supplementary materials. We have also fixed random seeds and specified hardware/software environments to facilitate consistent reproduction of our results.

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

# MAGREF: Masked Guidance for Any-Reference Video Generation with Subject Disentanglement

## Appendix

## A  PRELIMINARIES

In this section, we summarize the basic formulation of our generative framework. The core idea of video diffusion and flow-matching models is to construct a continuous transformation that maps gaussian noise into structured video data, conditioned on external signals such as text.

**Flow Matching.**  Instead of relying on a stochastic diffusion process, we employ flow matching Lipman et al. (2022), which defines a deterministic trajectory between the noise distribution and the target video distribution. Let $x_1 \in \mathbb{R}^{T \times C \times H \times W}$ denote a clean video, and $x_0 \sim \mathcal{N}(0, I)$ a random noise sample. For a timestep $t \in [0, 1]$, the interpolated state is written as:

$$x_t = t\,x_1 + (1 - t)\,x_0. \tag{8}$$

**Velocity Field.**  The dynamics of the trajectory are governed by its velocity, obtained by differentiating with respect to $t$:

$$v_t = \frac{dx_t}{dt} = x_1 - x_0. \tag{9}$$

**Training Objective.** The model, parameterized by $\theta$, is trained to approximate the true velocity with a predictor $u(x_t, y, t; \theta)$, where $y$ represents the conditioning signal. The loss function is defined as:

$$\mathcal{L}(\theta) = \mathbb{E}_{x_1, x_0 \sim \mathcal{N}(0, I),\, y,\, t \in [0,1]} \left[ \| u(x_t, y, t; \theta) - v_t \|_2^2 \right]. \tag{10}$$

During inference, an ODE solver integrates the estimated velocity field to evolve noise samples toward realistic videos, guided by the conditioning input $y$. To reduce the computational burden of high-dimensional video data, we employ a pretrained variational autoencoder (VAE) to map raw video sequences $X \in \mathbb{R}^{F \times C \times H \times W}$ into a compact latent representation $z_x \in \mathbb{R}^{f \times c \times h \times w}$. Here, $(F, C, H, W)$ and $(f, c, h, w)$ denote the frame, channel, and spatial dimensions before and after compression, respectively. This transformation preserves essential semantics while substantially reducing memory and computation cost.

# B    DATA CURATION PIPELINE

We present the detailed data curation pipeline used in our work, as illustrated in Figure 4. The pipeline is designed to construct a high-quality, large-scale training dataset in a systematic manner. It consists of four stages: (1) general filtering and captioning, (2) object processing and filtering, (3) face processing and filtering, and (4) cross-pair construction. Each stage progressively refines the raw data, ensuring both quality and diversity.

## B.1    GENERAL FILTERING AND CAPTIONING.

In the first stage, we aim to ensure a clean and diverse video corpus by segmenting raw training videos into high-quality clips. To achieve this, we apply scene change detection to break each video into multiple clips, denoted as $V_1, V_2, \ldots, V_n$. Unlike traditional text-to-video approaches, which focus on broad video content, our goal is to generate subject-centric video captions. To achieve this, we employ Qwen2.5-VL Bai et al. (2025), a large vision-language model, to describe the appearance and changes of the subject, while preserving key contextual elements of the video such as the environment and camera movements. The model generates captions $C_i$ that emphasize the subject's actions and visual transformations over time, while ensuring that environmental details and movement cues remain intact. We then evaluate the aesthetic quality and motion amplitude of each clip and discard any that do not meet the required standards, ensuring high-quality and subject-relevant training data.

## B.2    OBJECT PROCESSING AND FILTERING.

The second stage focuses on extracting and refining object-centric representations. For each filtered video clip $V_i$, we first extract candidate object labels from the generated captions $C_i$ using Qwen2.5-VL Bai et al. (2025). These labels represent the objects present in the video, such as "cat" or "dog," which serve as initial object candidates for further processing. Next, we apply GroundingDINO Liu et al. (2024), a grounding model that localizes each object by predicting bounding boxes in the frames. Let the bounding box for an object $k$ in video clip $V_i$ be represented as:

$$B_{i,k} = (x_{i,k}, y_{i,k}, w_{i,k}, h_{i,k}), \tag{11}$$

where $(x_{i,k}, y_{i,k})$ denotes the top-left corner of the bounding box, and $(w_{i,k}, h_{i,k})$ represent its width and height, respectively.

Once we have the bounding boxes, we use SAM2 Ravi et al. (2024) to segment these regions into object reference images $I_{i,k}^{\text{Obj}}$. The segmentation masks are generated by SAM2 and represent the precise boundaries of the detected objects. The corresponding reference image for an object $k$ in clip $V_i$ is denoted as:

$$I_{i,k}^{\text{Obj}} = \text{SAM2}(V_i, B_{i,k}), \tag{12}$$

where SAM2 is the segmentation model applied to the bounding box $B_{i,k}$ to generate an accurate segmentation mask for the object.

To ensure the reliability and accuracy of the object references, we further refine the segmentation masks by applying morphological operations, such as erosion and dilation, to smooth the object

boundaries and remove small noise. Let the refined mask be denoted as $\hat{M}_{i,k}$:

$$\hat{M}_{i,k} = \text{Morphology}(M_{i,k}), \tag{13}$$

where $M_{i,k}$ is the original segmentation mask, and $\hat{M}_{i,k}$ is the refined version obtained by applying the erosion and dilation operations. These operations help to eliminate small artifacts and improve the continuity of the object boundaries.

Additionally, we remove objects that are either too small or have abnormally irregular shapes, which are unlikely to provide useful features for training. This is done by using a size threshold $\theta_{\min}$ to discard small objects:

$$\text{if area}(M_{i,k}) < \theta_{\min}, \quad \text{remove } I_{i,k}^{\text{Obj}}, \tag{14}$$

where $\text{area}(M_{i,k})$ is the pixel area of the segmentation mask $M_{i,k}$, and $\theta_{\min}$ is the minimum size threshold. Objects that fall below this size are considered noise and discarded from the object references.

For objects that overlap with human faces in the scene, we apply Non-Maximum Suppression (NMS) to eliminate redundant object masks that may cause false positives or conflicts with human subjects. The overlap between two masks, $M_{i,k}$ (object mask) and $M_{i,\text{Face}}$ (face mask), is calculated using the Intersection-over-Union (IoU) metric:

$$\text{IoU}(M_{i,k}, M_{i,\text{Face}}) = \frac{|M_{i,k} \cap M_{i,\text{Face}}|}{|M_{i,k} \cup M_{i,\text{Face}}|}, \tag{15}$$

where $M_{i,k}$ and $M_{i,\text{Face}}$ are the object and face masks, respectively. If the IoU between the object mask and the human face mask exceeds a threshold of 0.25, we consider it a significant overlap and remove the object reference image $I_{i,k}^{\text{Obj}}$:

$$\text{if IoU}(M_{i,k}, M_{i,\text{Face}}) > 0.25, \quad \text{remove } I_{i,k}^{\text{Obj}}. \tag{16}$$

These steps ensure that the object references $I_{i,k}^{\text{Obj}}$ are high-quality, relevant, and suitable for use in subsequent stages of the pipeline. By removing small, noisy, or irrelevant objects and refining the segmentation results, we can better capture the objects that are most important for understanding the video content, thereby improving the overall quality of the training dataset.

## B.3 FACE PROCESSING AND FILTERING.

Human faces are a critical aspect of identity preservation in video data, especially for tasks involving consistent subject tracking or identity recognition. Therefore, we dedicate a separate stage for face-specific processing to ensure that face-related features are accurately extracted and maintained throughout the video clips.

We begin by using InsightFace[2], a state-of-the-art face recognition and analysis library, to detect faces across all frames in each video clip $V_i$, as well as in adjacent segments. For each frame, InsightFace detects multiple potential faces and extracts features that represent the face's unique identity. Each face is then embedded into a high-dimensional feature space, providing a robust representation of the identity. This embedding is used for identity assignment, allowing us to differentiate between different faces within the same or across clips.

In order to increase the reliability of the face detection process, we also estimate pose attributes for each detected face. Specifically, we calculate the yaw, pitch, and roll of each face, which represent its orientation in space. These pose attributes are important for distinguishing faces that may be tilted or viewed from uncommon angles. To improve robustness, we discard faces with extreme pose values or those detected at low quality (e.g., blurred or occluded faces), as they would introduce noise into the dataset.

For each unique identity, we rank the detected faces by two criteria: detection confidence and pose quality. The detection confidence is a measure of how certain the model is that the detected face is indeed a face, while the pose quality reflects how well the face's orientation aligns with standard frontal views. Faces with high detection confidence and optimal pose qualities are prioritized.

---

[2]https://github.com/deepinsight/insightface

To ensure balanced representation of identities across all frames, we uniformly sample 10 faces for each unique identity, avoiding the risk of over-representing any specific pose. These 10 selected faces are chosen to span the diversity of poses and qualities available within the clip, ensuring that we capture a wide range of possible face orientations while maintaining identity consistency.

The selected faces are then assembled into a set of human reference faces $I_i^{\text{Face}}$, which represents the set of cropped face reference images for each video clip $V_i$. Specifically, $I_i^{\text{Face}}$ contains the highest-scoring, frontal faces detected within the clip, ensuring that the final set of faces used for training is both diverse and consistent across different poses and quality levels.

We denote the set of all human reference faces across all video clips in the dataset as IFace, which represents the collection of all selected face reference images from the entire corpus:

$$\text{IFace} = \bigcup_i \left\{ I_{i,1}^{\text{Face}}, I_{i,2}^{\text{Face}}, \ldots, I_{i,k}^{\text{Face}} \right\}, \tag{17}$$

where $I_{i,k}^{\text{Face}}$ represents the $k$-th highest-scoring frontal face from the $i$-th video clip. Each $i$ corresponds to a different video clip, and the set $\{I_{i,1}^{\text{Face}}, \ldots, I_{i,k}^{\text{Face}}\}$ contains the highest-scoring frontal faces detected in clip $i$, ensuring high confidence in identity consistency. This set ensures that the model can consistently reference and learn from human faces that are not only representative of the subject but also exhibit optimal frontal orientation for better identity recognition and pose estimation.

Formally, each curated training sample after Stage 3 processing is defined as:

$$\mathcal{R}_i = \left\{ V_i,\ C_i,\ I_i^{\text{Face}},\ I_{i,1}^{\text{Obj}},\ I_{i,2}^{\text{Obj}},\ \ldots,\ I_{i,k}^{\text{Obj}} \right\}, \tag{18}$$

where $V_i$ denotes the ground-truth video clip, $C_i$ represents the text caption, $I_i^{\text{Face}}$ denotes the cropped face reference, and $I_{i,j}^{\text{Obj}}$ corresponds to the object references. This structured representation ensures that each training sample is aligned with the relevant video content, captions, faces, and objects, enabling robust model training.

## B.4 Cross-Pair Data Construction.

In Stage 4, we focus on addressing the issue of copy-paste artifacts, which often arise when the model generates videos where the poses and orientations of subjects remain overly consistent with those in the reference images. This stage aims to enhance the diversity of the multi-subject dataset and improve the model's ability to generalize across varied visual scenarios. By leveraging an image generation model, we generate transformed variants of both face and object references, reducing the risk of overfitting to specific, static object-background pairings.

For each face reference $I_i^{\text{Face}}$ and each object reference $I_{i,j}^{\text{Obj}}$ obtained in Stages 2 and 3, the image generation model produces augmented counterparts $I_i^{\text{Face}'}$ and $I_{i,j}^{\text{Obj}'}$ with variations in pose, appearance, and context. These transformations are designed to simulate a broader range of real-world conditions, ensuring that the model is not simply learning from unaltered, static pairings of faces, objects, and backgrounds. These variations help mitigate the potential for the model to learn artifacts related to repetitive patterns, such as those caused by directly copying objects onto backgrounds. In addition to the face and object transformations, background images are also augmented to further enrich the reference set.

The goal of this augmentation process is to create a dataset where the foreground (face and object) and background are less rigidly linked, preventing the model from relying too heavily on repetitive object-background pairings and thereby reducing the risk of copy-paste artifacts.

Formally, each training sample after Stage 4 is defined as:

$$\mathcal{R}_i = \left\{ V_i,\ C_i,\ (I_i^{\text{Face}}, I_i^{\text{Face}'}), (I_{i,1}^{\text{Obj}},\ I_{i,1}^{\text{Obj}'}),\ \ldots,\ (I_{i,k}^{\text{Obj}},\ I_{i,k}^{\text{Obj}'}),\ I_i^{\text{Bg}} \right\}, \tag{19}$$

where $(I_i^{\text{Face}}, I_i^{\text{Face}'})$ are the original and transformed face references, $(I_{i,j}^{\text{Obj}}, I_{i,j}^{\text{Obj}'})$ represent the object-variant pairs, and $I_i^{\text{Bg}}$ denotes the background reference. The training sample $\mathcal{R}_i$ now includes both the original and transformed face and object references, along with a corresponding background, resulting in a more diverse and less repetitive training dataset.

By integrating these augmented references into the training process, we ensure that the model not only learns from high-quality, subject-centric data but also gains the ability to generalize effectively across a wide variety of visual contexts. This strategy significantly reduces the occurrence of copy-paste artifacts, leading to more natural and realistic interactions in generated videos. Together, these four stages form a systematic pipeline that transforms raw, noisy video data into high-quality, semantically aligned training samples, essential for scalable and controllable any-reference video generation.

# C  EXPERIMENT SETTINGS

## C.1  EVALUATION BENCHMARK

Existing benchmarks for any-reference video generation have notable limitations, particularly in assessing the flexibility and robustness of generative models across a wide range of complex scenarios. To address this gap, we propose a systematic and task-specific benchmark designed to comprehensively evaluate our video generation framework in both single-ID and multi-subject settings. This benchmark consists of 120 subject-text pairs, divided into two primary categories: single-ID and multi-subject. The single-ID group includes 60 test cases, each involving a single ID reference image, while the multi-subject group encompasses 60 cases with varying complexities, such as two-person, three-person, and mixed scenarios, including human-object-background compositions.

A subset of the benchmark is adapted from existing datasets such as ConsisID Yuan et al. (2024), OpenS2V Yuan et al. (2025), and A2-Bench Fei et al. (2025), ensuring a consistent foundation for comparison. The remaining cases are carefully curated to guarantee comprehensive coverage across diverse subject types, background settings, and interaction dynamics. Each test case consists of no more than three reference images, accompanied by a natural language prompt designed to maintain high aesthetic quality and semantic alignment. This controlled structure ensures consistent difficulty across the benchmark, allowing for a detailed and rigorous evaluation of the generative model's performance.

The diversity of the benchmark is integral to its design, incorporating varying subject appearances, prompt lengths, and compositional arrangements. This enables a fine-grained evaluation of the model's ability to synthesize coherent and diverse videos, accounting for a broad spectrum of visual and semantic complexity. By incorporating real-world elements such as varying background contexts and dynamic subject interactions, the benchmark provides a robust testbed for evaluating the model's capacity to generate realistic, high-fidelity videos under challenging conditions. This approach not only ensures a comprehensive assessment of the model's generative capabilities but also highlights its potential for generalization across a variety of complex scenarios.

## C.2  EVALUATION MODELS

We evaluate a representative set of mainstream proprietary and open-source models for the any-reference video generation task, comprising 4 proprietary and 8 open-source models. The proprietary models include Hailuo (2025), Pika (2025), Vidu (2025), and Kling (2025). Among these, Hailuo is evaluated in the single-ID setting, whereas Pika, Vidu, and Kling are evaluated on both single-ID and multi-subject tasks. For open-source baselines, ConsisID Yuan et al. (2024), EchoVideo Wei et al. (2025), FantasyID Zhang et al. (2025), Concat-ID Zhong et al. (2025) , and HunyuanCustom Hu et al. (2025) are used for single-ID evaluation. SkyReels-A2 Fei et al. (2025), Phantom Liu et al. (2025), and VACE Jiang et al. (2025) are evaluated on both single-ID and multi-subject tasks. The detailed evaluation protocols and configuration specifics are provided below.

For Hailuo, we use the official S2V function of Hailuo-S2V-01 with default settings, generating a 5-second video (141 frames) at a resolution of $1280 \times 720$ and a frame rate of 25 fps. For Pika, we utilize the official Pika 2.1 with default parameter settings, producing a 5-second video (121 frames) at a resolution of $1920 \times 1080$ and a frame rate of 24 fps, which allows for a comprehensive assessment of the model's performance in generating high-resolution videos. For Vidu, we use Vidu 2.0's *character-to-video* function with default settings in *turbo* mode, generating a 4-second clip (65 frames) at 16 fps with a spatial resolution of $704 \times 396$ and automatic motion amplitude. For Kling, we employ the official Kling 1.6 with default settings, producing a 5-second video (153 frames) at a resolution of $1280 \times 720$ and a frame rate of 30 fps, enabling an in-depth evaluation of the model's performance across varying visual and semantic contexts.

For open-source single-id evaluation, we use the official code and models for ConsisID, EchoVideo, FantasyID, Concat-ID, and HunyuanCustom, maintaining the original settings. For ConsisID, videos are generated at a spatial resolution of $720 \times 480$ with a frame rate of 8 fps, yielding a duration of 6 seconds (49 frames). EchoVideo generates 3-second videos (49 frames) at a resolution of $848 \times 480$ and a frame rate of 16 fps. FantasyID generates 6-second videos (49 frames) at $720 \times 480$ and 8 fps. Concat-ID generates 5-second videos (81 frames) at $832 \times 480$ and 16 fps for the Wan-AdaLN version. Lastly, HunyuanCustom generates 5-second videos (129 frames) at a resolution of $1280 \times 720$ and 25 fps. Each setup ensures consistency in video generation while varying resolution and frame rate across the models for effective comparison.

For open-source multi-subject evaluation, we use the official code and models for SkyReels-A2, Phantom, and VACE, maintaining the original settings. For SkyReels-A2, we employ the A2-Wan2.1-14B-Preview model, generating 5-second videos (81 frames) at a resolution of $832 \times 480$ and a frame rate of 16 fps. In Phantom, we use the Phantom-Wan-14B model to generate 5-second videos. For VACE, we use the VACE-Wan2.1-14B to generate 5-second videos (81 frames) at $1080 \times 720$ and 16 fps. Each setup ensures consistent video length, frame rate, and model performance across the three models, allowing for effective comparison.

## C.3 DETAILED EVALUATION METRICS.

For evaluation, we consider both single-ID and multi-subject settings to comprehensively assess model performance. The following sections provide a detailed explanation of the six evaluation metrics used in this study, focusing on their practical relevance in any-reference video generation tasks.

**ID-Sim (Identity Similarity)**  The ID-Sim metric measures the consistency of the human's identity across video frames. This is done by calculating the cosine similarity between face embeddings extracted from each frame of the video using a pretrained face recognition model, ArcFace Deng et al. (2019). To ensure a representative evaluation, we select frames at regular intervals (every 16th frame) and exclude frames where no face is detected. The cosine similarity $\text{sim}(A, B)$ between two face embeddings $A$ and $B$ is computed as:

$$\text{sim}(A, B) = \frac{A \cdot B}{\|A\|\|B\|}, \tag{20}$$

where $A \cdot B$ is the dot product of the embeddings and $\|A\|$ and $\|B\|$ are their respective Euclidean norms. A higher cosine similarity value indicates better identity preservation across frames, meaning that the model maintains the subject's appearance and characteristics consistently throughout the video.

**Aesthetic Score**  The Aesthetic Score christophschuhmann (2024) evaluates the visual quality of the generated video based on human perceptual preferences. This metric is derived from a learned aesthetic prediction model, which is trained on a large dataset of high-quality images capturing subjective factors such as color harmony, sharpness, and overall composition. The aesthetic score $S_{\text{aesthetic}}$ for the entire video is calculated as the average of the frame-wise aesthetic scores $S_{\text{aesthetic}}(I_t) = f(I_t)$, where $f(I_t)$ represents the learned function that outputs the visual appeal of each frame. The overall score is then given by:

$$S_{\text{aesthetic}} = \frac{1}{N} \sum_{t=1}^{N} f(I_t), \tag{21}$$

where $N$ is the total number of frames in the video.

**Motion Smoothness**  To evaluate the fluidity of motion in the generated video, we employ the Motion Smoothness metric Wu et al. (2023). This metric measures the temporal coherence of movement between consecutive frames, which is essential for ensuring that motion transitions smoothly and naturally, without abrupt changes or artifacts. It is crucial for maintaining the realism and continuity of dynamic actions within the video.

**GmeScore** The GmeScore Zhang et al. (2024) is used to evaluate the semantic alignment between the generated video and the input text. Traditional models such as CLIP and BLIP are often used for text-to-image or text-to-video relevance but are limited by short token lengths (usually 77 tokens), which makes them unsuitable for handling long-form text prompts typical in DiT-based video generation models. GmeScore is based on a vision-language alignment model fine-tuned on Qwen2-VL and is capable of processing longer and more complex text descriptions.

**Subj-Sim (Subject Similarity)** The Subj-Sim metric assesses the consistency of the subject across video frames. For each video, we sample frames at equal intervals and extract the regions corresponding to the subject in both the generated video and the ground-truth (GT) images using segmentation models like GroundingDINO Liu et al. (2024) and SAM2 Ravi et al. (2024). The embeddings for both the GT subject and the video frame subject are obtained using the DINO model. The cosine similarity $\mathrm{sim}(S_i, S_{\mathrm{gt}})$ between the embeddings of the subject regions $S_i$ from the video frames and the ground-truth subject $S_{\mathrm{gt}}$ is calculated for each frame $i$:

$$\mathrm{sim}(S_i, S_{\mathrm{gt}}) = \frac{S_i \cdot S_{\mathrm{gt}}}{\|S_i\| \|S_{\mathrm{gt}}\|}, \tag{22}$$

where $S_i$ and $S_{\mathrm{gt}}$ are the embeddings of the subject in the $i$-th video frame and the GT subject, respectively. The average similarity score $\overline{S_{\mathrm{subj}}}$ is then computed by averaging the cosine similarities over all sampled frames:

$$\overline{S_{\mathrm{subj}}} = \frac{1}{N} \sum_{i=1}^{N} \mathrm{sim}(S_i, S_{\mathrm{gt}}), \tag{23}$$

where $N$ is the total number of frames sampled. Higher similarity values indicate better consistency in the subject's appearance across frames.

**Bg-Sim (Background Similarity)** The Bg-Sim metric evaluates the consistency of the background across video frames. Similar to Subj-Sim, we calculate the similarity between the background of the inpainted video frames and the ground-truth background by sampling frames at equal intervals. The inpainting model Podell et al. (2023) is used to reconstruct missing or altered regions of the background in the video. The DINO model is used to extract embeddings for both the inpainted background and the ground-truth background. The cosine similarity $R_{\mathrm{bg}}$ between the embeddings of the inpainted background $B_i$ and the ground-truth background $B_{\mathrm{gt}}$ for each frame $i$ is calculated as:

$$R_{\mathrm{bg}} = \frac{B_i \cdot B_{\mathrm{gt}}}{\|B_i\| \|B_{\mathrm{gt}}\|}, \tag{24}$$

where $B_i$ and $B_{\mathrm{gt}}$ are the embeddings of the inpainted background and the ground-truth background for the $i$-th frame, respectively. The average background similarity $\overline{S_{\mathrm{bg}}}$ is computed by averaging the cosine similarities over all sampled frames:

$$\overline{S_{\mathrm{bg}}} = \frac{1}{N} \sum_{i=1}^{N} R_{\mathrm{bg}}, \tag{25}$$

where $N$ is the total number of frames sampled. Higher background similarity values indicate that the background remains consistent and realistic across frames.

## D  ADDITIONAL ABLATION RESULTS

### D.1  ABLATION DETAILS ON MASKED GUIDANCE

In this section, we conduct a detailed evaluation of the two central components of masked guidance, the region-aware masking mechanism and the pixel-wise channel concatenation mechanism, and provide an in-depth analysis of their effectiveness.

**Region-aware masking mechanism.** The region-aware masking mechanism is designed to accommodate a variable number of reference images in a spatially adaptive and content-aware manner. Rather than relying on a fixed concatenation strategy, it selectively modulates the visible regions

of each reference image during training, enabling the model to dynamically allocate attention to semantically meaningful areas. This fine-grained mechanism aligns more closely with the natural variability of multi-subject and multi-object scenes, where different references may occupy distinct spatial regions or contribute unevenly across time.

To illustrate its effect, we compare two masking strategies: a fine-grained region-aware masking mechanism (top of Figure 6) and a coarse-grained vanilla masking mechanism, which follows the design of SkyReels-A2 Fei et al. (2025) (bottom of Figure 6). The vanilla approach concatenates reference images directly along the channel dimension, ignoring spatial locality. As shown in Figure 6(b) and (c), this naïve strategy often causes frame-level inconsistencies and identity drift, particularly during long video synthesis. Even after discarding the initial warm-up frames, subsequent generations frequently degrade in visual quality, leading to unstable motion and the gradual loss of subject fidelity. These issues indicate that coarse channel concatenation combined with uniform masking introduces strong interference, which undermines temporal coherence and hinders the stable inheritance of subject identity.

In contrast, the region-aware masking mechanism explicitly regulates the contribution of each reference image across both space and time. By masking irrelevant or redundant regions and preserving only task-relevant cues, the model avoids channel-level entanglement and significantly reduces cross-subject interference. This allows the generator to better exploit fine-grained visual information, while simultaneously maintaining consistency with the I2V training paradigm. As a result, the generated videos exhibit sharper details, smoother motion dynamics, and more faithful preservation of subject identity, even under long-horizon generation settings. Overall, this ablation study highlights that spatially adaptive region-aware masking is crucial for stabilizing training, reducing identity drift, and improving the perceptual quality of any-reference video generation.

**Pixel-wise channel concatenation mechanism.** We perform ablation experiments to compare two strategies for integrating reference images: the proposed pixel-wise channel concatenation mechanism and the token-wise concatenation mechanism commonly adopted in prior work Hu et al. (2025); Liu et al. (2025). As shown in Figure 6(d), our pixel-wise channel concatenation consistently demonstrates superior identity preservation, especially in reconstructing fine-grained facial structures and subtle appearance cues. By embedding reference images directly into spatially aligned feature channels, the model receives strong supervision signals that are tightly coupled with the spatial layout of the generated frames.

In contrast, the token-wise concatenation approach treats reference images as additional tokens that are injected into the transformer input sequence. In this setting, the model relies entirely on self-attention layers to extract and propagate identity-related information. Such indirect encoding weakens the supervision of identity cues during training, since identity information is scattered across tokens and more prone to diffusion. As illustrated in Figures 6(e) and (f), this often results in inconsistencies in subject appearance, such as blurred facial features, unstable textures, or even identity drift over longer generations.

These shortcomings become even more pronounced when the model encounters out-of-domain reference images, where the distributional gap between training data and unseen references challenges its generalization ability. Under token-wise concatenation, the model struggles to robustly transfer identity cues from such references, frequently producing distorted or mismatched identities. In contrast, pixel-wise concatenation leverages spatially grounded and semantically rich features that anchor identity information more effectively, thereby reducing failure cases in out-of-domain scenarios. Overall, these results highlight the advantages of our design: by directly embedding reference cues in pixel-aligned representations, our approach significantly improves both in-domain fidelity and out-of-domain generalization in any-reference video generation.

## D.2 ABLATION DETAILS ON REFERENCE IMAGE SCALABILITY

To further evaluate the scalability of our approach with respect to reference images, we perform a comprehensive qualitative experiment. As shown in Figure 7, we provide qualitative examples that demonstrate how the model performs with reference images of varying scales. These examples illustrate how the subject and style information is preserved and transferred effectively across different sizes, even when the reference images are significantly reduced. The results clearly indicate that the

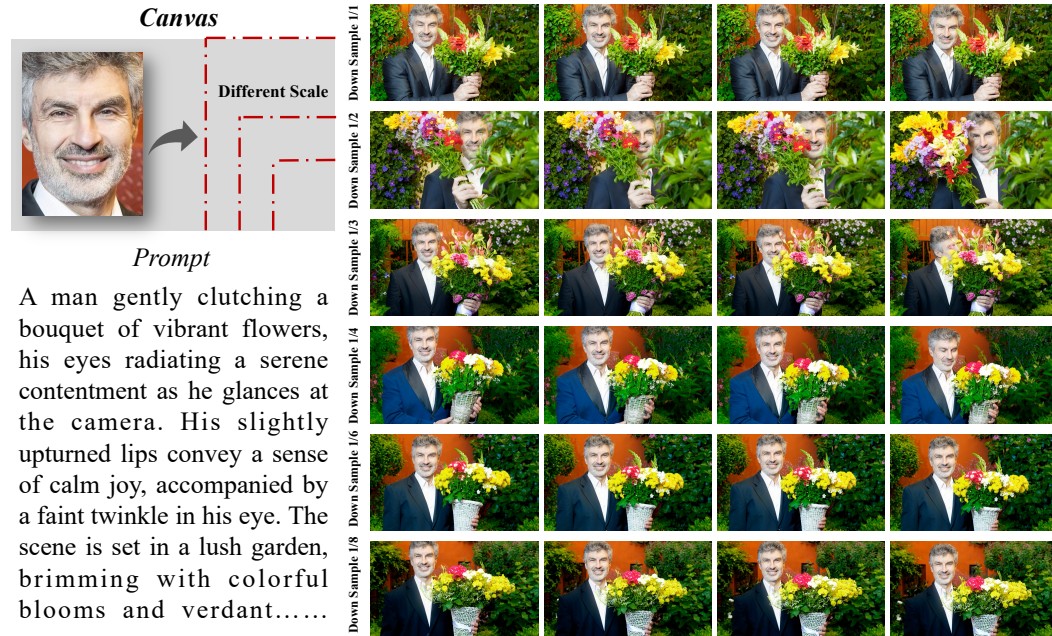

Figure 7: **Qualitative examples showing how the model handles reference images of different scales.** The model maintains consistency in the generated output, even when the reference image is significantly reduced.

model maintains consistency in the generated output, regardless of the reference image's size. This observation suggests that our approach can handle up to 8 reference images simultaneously without a significant loss of quality or detail. Furthermore, we find that the overall effect does not exhibit substantial variations with changes in the size of the reference images. These findings demonstrate the robustness and flexibility of our method in managing a range of image scales, making it scalable for various applications.

### D.3 ABLATION ON SUBJECT DISENTANGLEMENT MECHANISM

Figure 9 presents more qualitative results of the ablation study on the Subject Disentanglement (SD) mechanism. When SD is removed, we observe severe entanglement between different subjects, such as blending of facial features, inconsistent appearances across frames, and failure to maintain distinct identities in multi-subject scenarios. For example, in the first row, the absence of SD causes the doctor and patient to gradually lose their unique characteristics, leading to identity drift. Similarly, in the second case, the two individuals in the selfie scene show visual confusion, with faces and attributes becoming entangled over time. The third case demonstrates that in human–animal interactions, the model without SD not only fails to preserve subject identities but also hallucinates an additional dog, indicating entanglement and instability in multi-subject scenarios. By contrast, our full model with SD effectively disentangles subjects, maintains identity fidelity, and produces temporally coherent results across diverse scenarios. These results highlight the importance of the SD mechanism for handling complex any-reference generation tasks.

## E MORE QUALITATIVE RESULTS

### E.1 MORE RESULTS OF MAGREF

We provide additional qualitative results of our method in Figures 10–14, which further demonstrate the effectiveness of MAGREF in synthesizing coherent videos from paired text prompts and reference images. Our model consistently preserves the distinct visual attributes of the provided references while faithfully following the input text conditions.

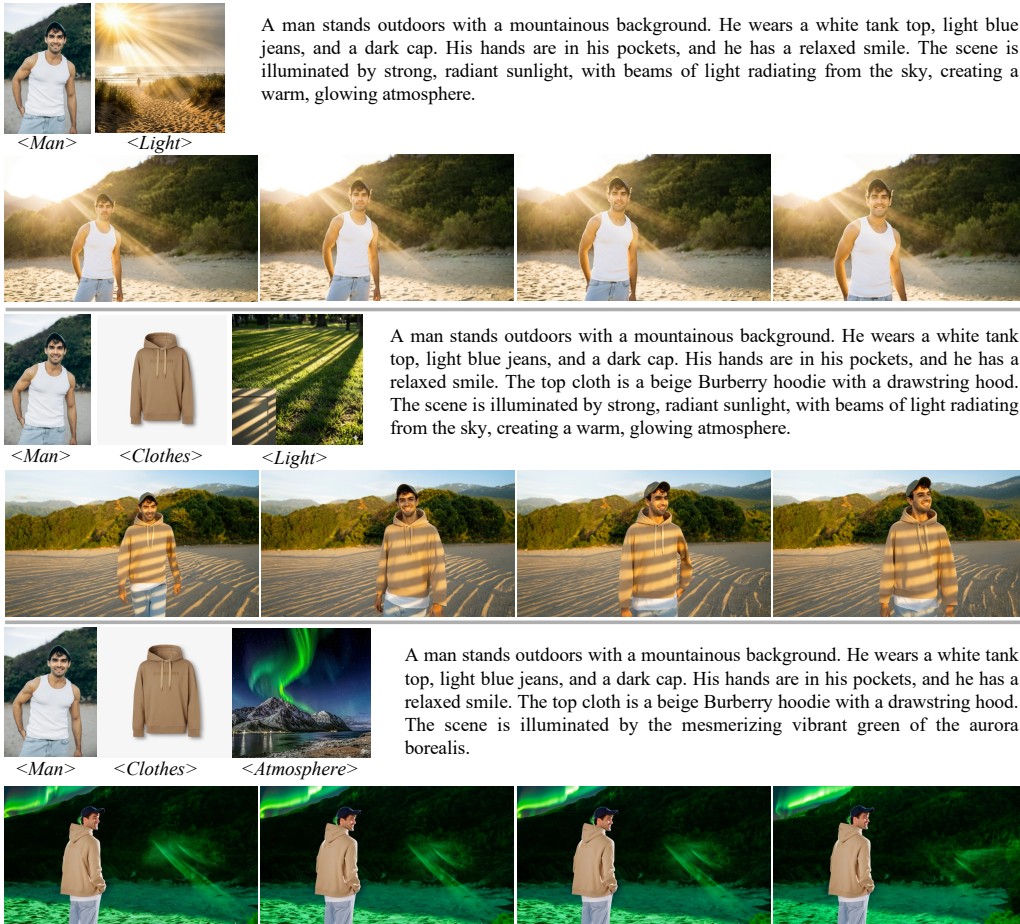

Figure 8: **More qualitative experiments exploring global element customization in video generation.** Qualitative examples demonstrate that our method can flexibly adapt global attributes like lighting and environmental atmosphere in a zero-shot manner. This highlights the potential of the method to control global features such as lighting and atmosphere without sacrificing subject identity.

Figure 10 highlights human–object compositions involving accessories such as bags, rings, and necklaces. Figure 11 extends this to glasses, while Figure 12 shows results with clothing such as blouses, polo shirts, hoodies, jackets, hats, and sweaters. Figure 13 illustrates single-ID cases, highlighting identity preservation across generated frames. Finally, Figure 14 demonstrates multi-subject scenarios, including persons, animals, and scenes.

Together, these results confirm that our model generalizes well across accessories, glasses, clothing, single-ID, and multi-subject composition tasks, accurately capturing interactions between people, objects, and environments while generating contextually appropriate and visually compelling videos.

## E.2    FAILURE CASES VISUALIZATION

Although our method demonstrates strong overall performance, certain failure cases still arise in specific scenarios (see 15). One key challenge is the scarcity of high-quality data that effectively captures complex subject interactions, which limits the model's ability to generalize, particularly in scenes involving multiple subjects or intricate subject-object dynamics. As a result, the model may struggle to maintain subject consistency and coherence when handling such interactions. Additionally, the current foundation models are insufficient in modeling physical laws, leading to unrealistic phenomena in some scenarios. For example, when large-scale motions are involved, the model

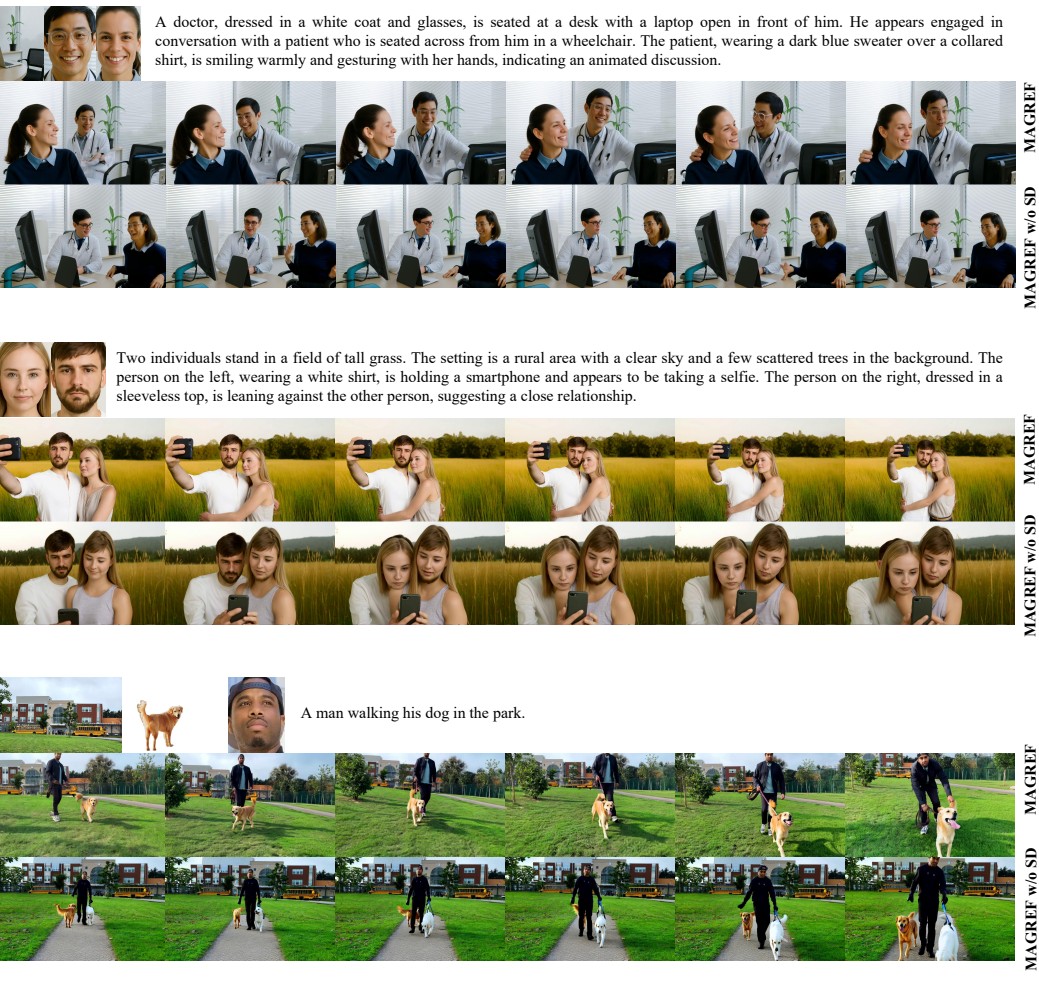

Figure 9: **More qualitative comparison for the ablation on Subject Disentanglement (SD).** The proposed MAGREF preserves subject identities and prevents entanglement across different scenes, while the variant without SD (MAGREF w/o SD) exhibits identity drift, blending, and loss of consistency when multiple human or animal subjects appear simultaneously.

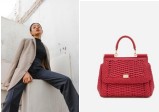 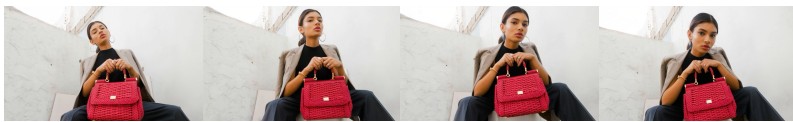

A young woman sitting against a weathered wall with a serious expression. She is wearing a beige blazer over a black high-neck top and dark pinstripe pants. She wears a red woven handbag with gold hardware.

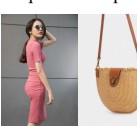 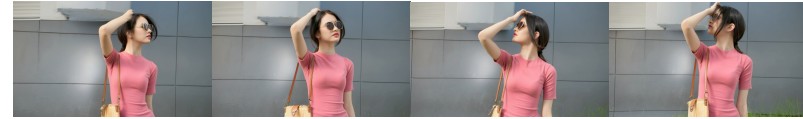

A young woman stands against a gray wall, wearing a fitted pink dress with short sleeves and a round neckline. Her hand rests on her head. She wears a natural-colored woven bag with a brown leather strap over her shoulder.

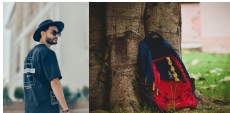 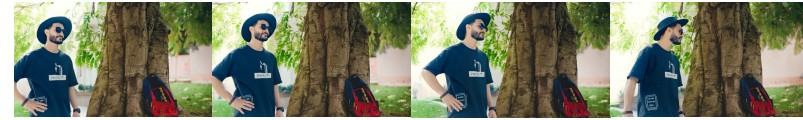

A man stands outdoors, leaning against a tree trunk. He wears a black hat, sunglasses, and a dark blue T-shirt, and he looks off to the side with a serious expression. A red and blue backpack leans against the tree next to him.

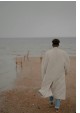 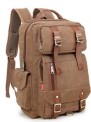 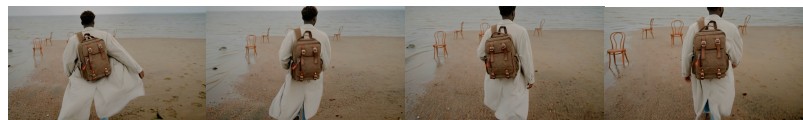

A man walking barefoot on a beach, wearing a beige trench coat and blue jeans. He is walking towards the ocean, with three orange chairs placed near the water. The bag is a brown canvas backpack with multiple pockets and straps.

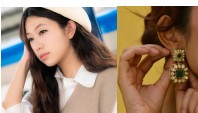 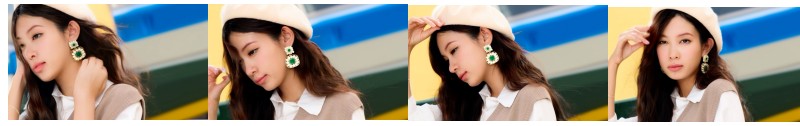

A close-up portrait of a young woman's face and upper body. She is wearing a beige beret, a white shirt with a button-down collar, and a beige cardigan. She is wearing large, dangling earrings with a floral design and green accents.

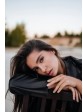 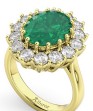 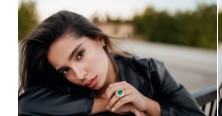 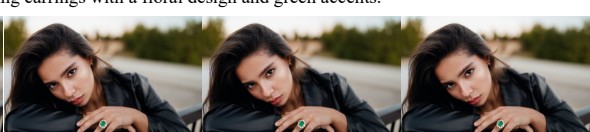

A close-up portrait of a young woman leaning on a railing. The woman is looking directly at the camera with a neutral expression. The woman wears a gold ring with a large emerald stone surrounded by smaller diamonds.

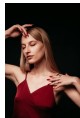 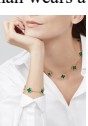 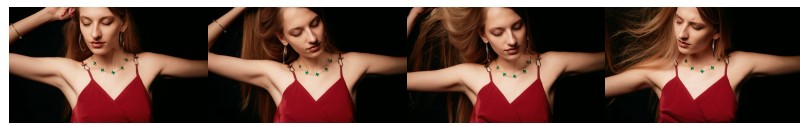

A young woman stands against a dark background, her eyes closed as she gently touches her hair with one hand while the other rests on her chest. The necklace, featuring green clover-shaped pendants, is worn around her neck.

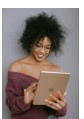 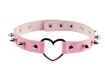 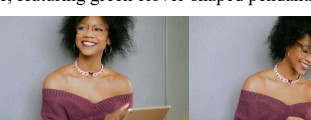 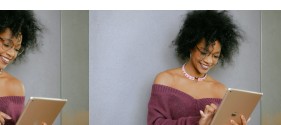 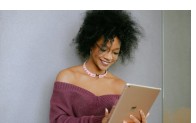

A young woman stands against a gray speckled wall, smiling while holding an iPad. Her curly hair frames her face as she looks down at the tablet. The necklace is a pink choker with silver spikes and a heart-shaped cutout.

Figure 10: **Qualitative results on any-reference human and object composition.** Each row conditions on two reference images: a human reference on the left and an object reference on the right (bag, ring, or necklace). Our model supports arbitrary pairings between humans and accessories, reliably identifies the intended object, even when the object reference is cluttered or contains distractors, and faithfully follows the text prompt, producing effects akin to video editing.

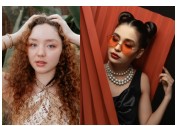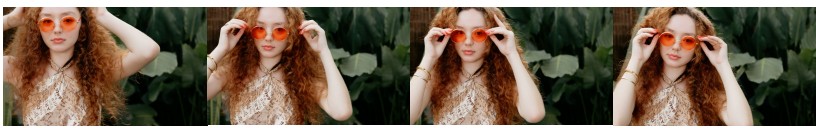

A close-up portrait of a young woman with curly red hair, wearing a black strapless top with a pearl necklace. She is holding her hair back with one hand and has a gold bracelet on her wrist. The woman is wearing orange-tinted sunglasses with a gold frame.

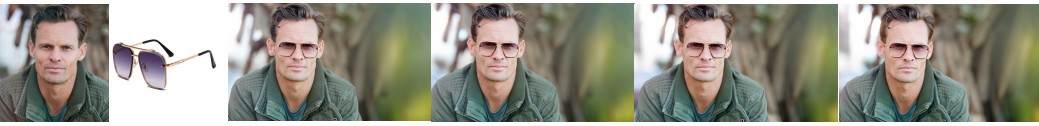

A close-up portrait of a man's face and upper body. He is wearing a green jacket with a high collar and a zipper. His hair is styled neatly, and he has a serious expression on his face. The man is wearing gold-framed sunglasses with gradient lenses.

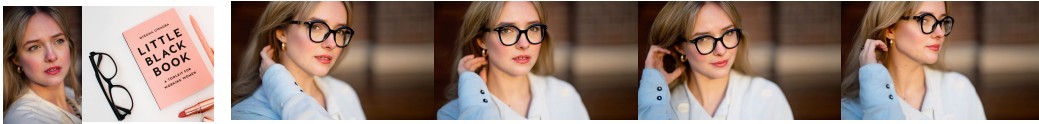

A close-up portrait of a young woman's face and upper body. Her hair is styled in loose waves, and she is wearing large hoop earrings. The woman is looking directly at the camera with a neutral expression. She wears black-framed glasses on her face.

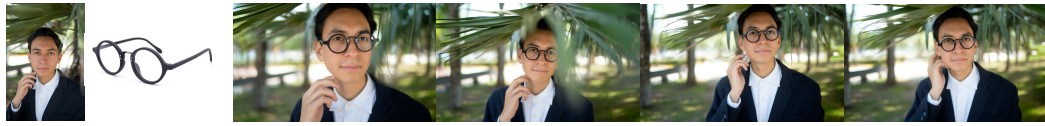

A close-up portrait of a man's face and upper body. He is wearing a green jacket with a high collar and a zipper. He has a serious expression on his face. The sunglasses have a sleek, modern design with a gunmetal frame and dark lenses.

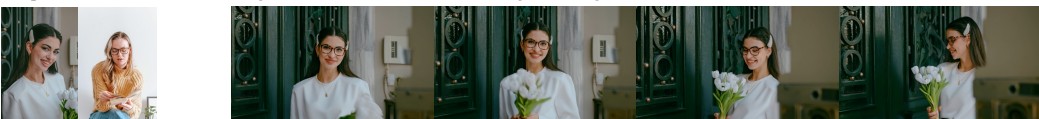

A young woman stands in front of a green door with a vintage camera in the background. She is wearing a white long-sleeve shirt. She holds a bouquet of white tulips and smiles warmly. She wears glasses with tortoiseshell frames.

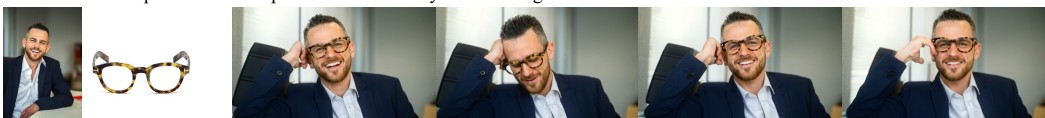

A man sitting at a table with his arm resting on it, smiling with a confident expression. He is wearing a dark blue suit jacket over a white shirt with the collar slightly open. The glasses have a tortoiseshell pattern with a gold logo on the temples.

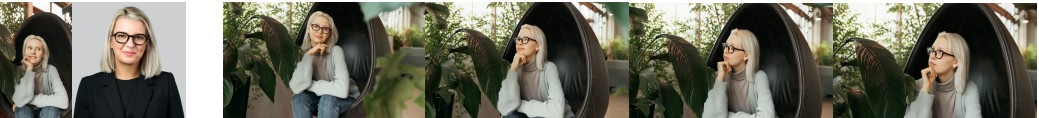

A young woman sitting in a black leather chair with green plants in the background. She is wearing black glasses with a thin gold earring in her left ear. The woman is resting her chin on her hand, looking slightly to the side with a thoughtful expression.



A close-up portrait of a man's upper body. The man is looking directly at the camera with a slight smile. The background is plain white. The man is wearing a pair of matte black Holbrook RX glasses.

Figure 11: **Qualitative results on any-reference human and glasses composition.** Each row conditions on two reference images: a human reference on the left and a glasses reference on the right. Our model supports arbitrary pairings between humans and eyewear, reliably identifies the intended glasses even when the reference image is cluttered or contains distractors, and faithfully follows the text prompt, producing effects similar to video editing.

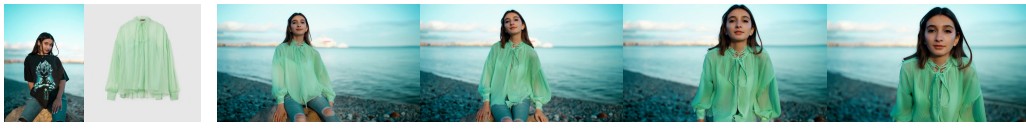

A young woman sitting on rocks by the sea, wearing a mint green sheer blouse with a high neckline and long sleeves. The blouse has a ruffled collar and a tie at the neck. The background is a serene blue ocean under a clear sky.

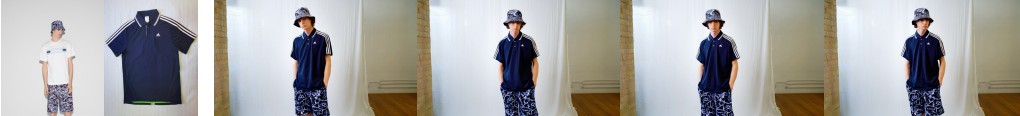

A young man standing in a studio with a white background. He is wearing a navy blue polo shirt with white stripes on the shoulders and sleeves.He has a relaxed pose with his hands in his pockets…

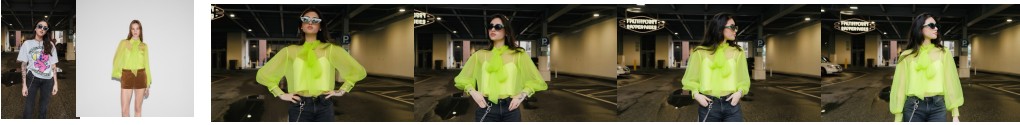

A young woman stands in an urban parking garage, wearing a neon green sheer blouse with a bow at the neckline. She has long dark hair, sunglasses, and tattoos on her arms. She wears high-waisted jeans with a chain accessory.

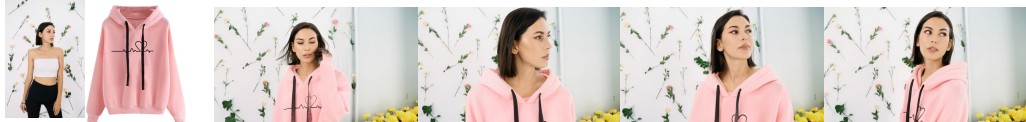

A young woman stands against a backdrop of flowers, wearing a pink hoodie with a heart and heartbeat design. She has a short bob hairstyle and is looking off to the side with a neutral expression.

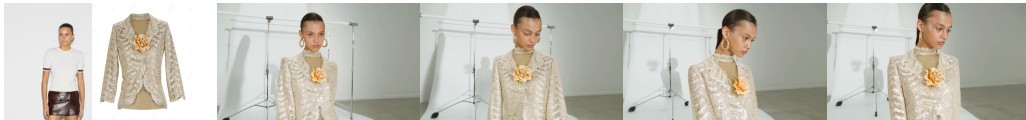

A young woman stands in a studio with a white background. She is wearing a lace jacket with long sleeves and a floral brooch at the center. The jacket is light beige with intricate patterns. Her hair is pulled back, and she wears gold hoop earrings.

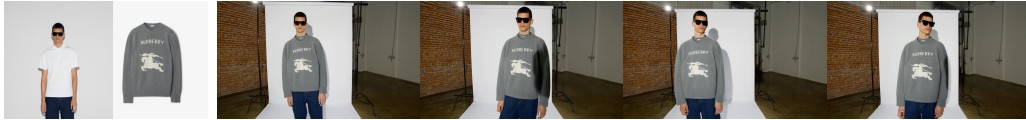

A young man standing in a studio with a white background. He is wearing a gray Burberry sweater with a white logo and a running horse design. The sweater has a round neckline and long sleeves. The man is also wearing dark sunglasses and blue jeans.

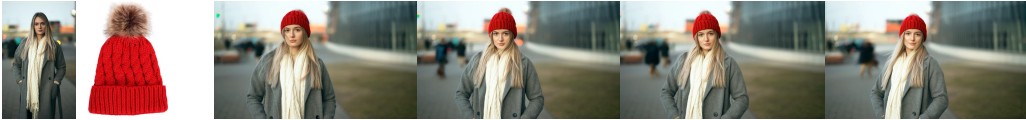

A young woman stands outdoors with a blurred cityscape in the background. Her hands are tucked into the coat pockets, and she looks directly at the camera with a neutral expression. She wears a red knitted hat with a large fur pom-pom on top.

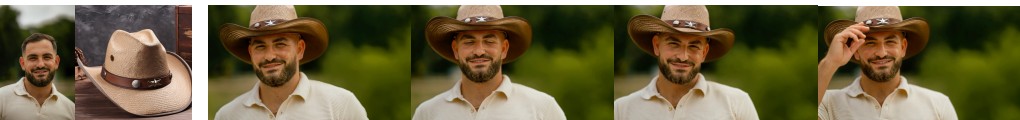

A close-up portrait of a man's face and upper body. The man is smiling slightly, with his eyes looking directly at the camera. The man is wearing a cowboy hat with a wide brim, a brown band, and decorative metal accents.

Figure 12: **Qualitative results on any-reference human and clothing composition.** Each row conditions on two reference images: a human reference on the left and a clothing reference on the right (e.g., blouse, polo shirt, hoodie, jacket, hat or sweater). Our model supports arbitrary pairings between humans and garments, reliably identifies the intended clothing item even when the reference is cluttered or includes distractors, and faithfully follows the text prompt, producing effects similar to video editing.

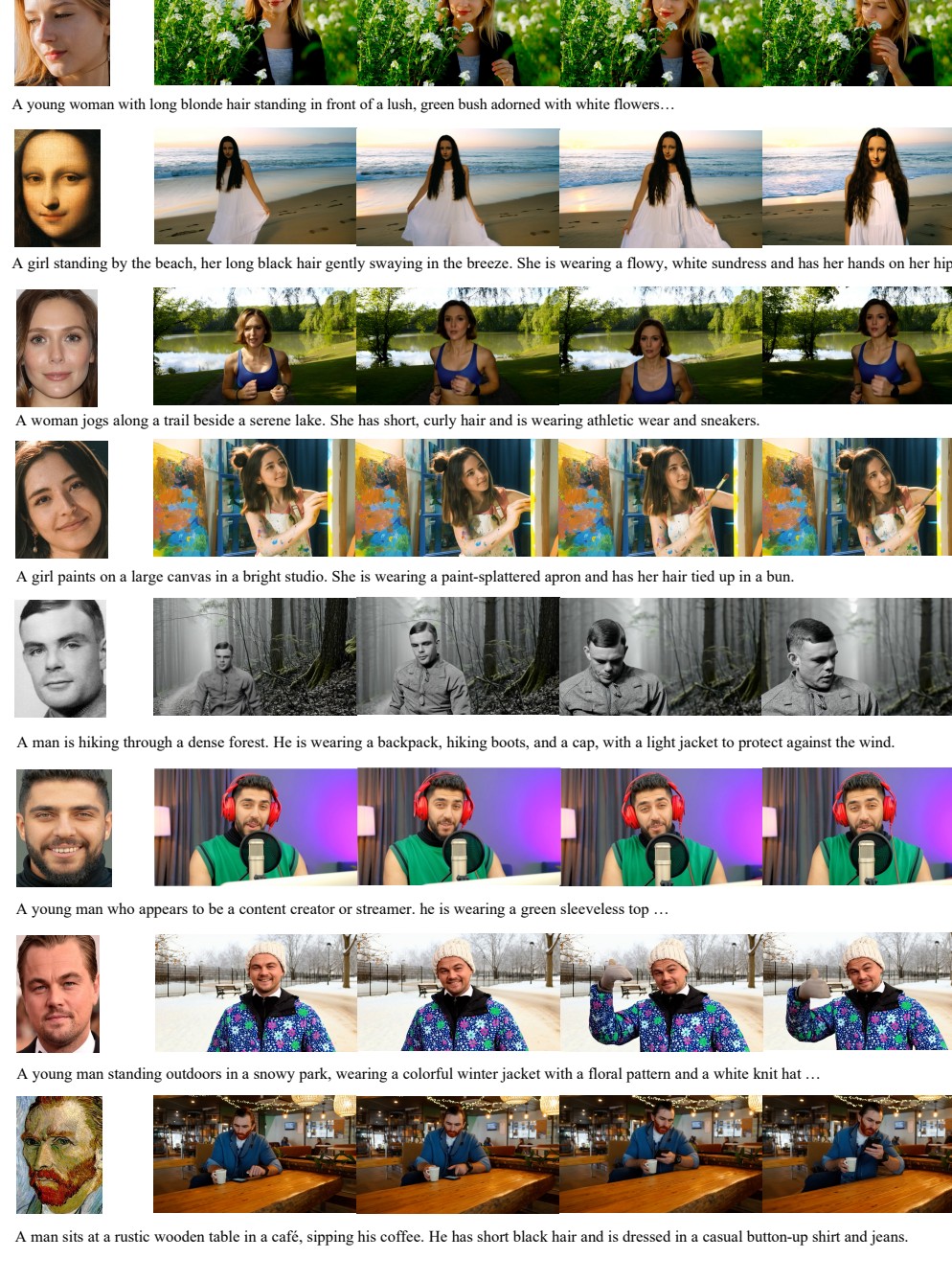

Figure 13: **Qualitative evaluation results of our method on test cases involving a single ID.** Our model consistently generates videos that maintain the subject's identity while accurately following the input text prompt.

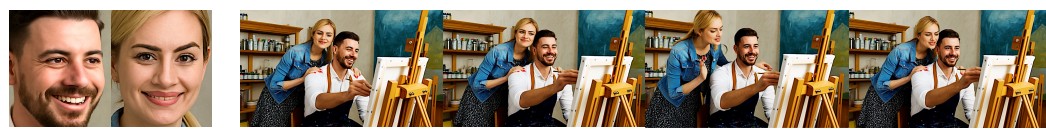

The video features two individuals, a man and a woman, dancing against a bright yellow background. Both are dressed casually; the man wears a red and black plaid shirt over a white t-shirt, while the woman is in a green button-up shirt layered over a white top.

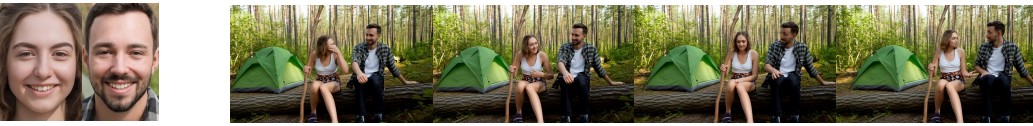

The video opens with a serene outdoor setting in a forest, featuring a green tent and two individuals seated on a log. The scene is set during the daytime, with clear weather and sunlight filtering through the trees. The individuals are dressed casually, with one person wearing a white tank top and patterned shorts, and the other in a plaid shirt and dark pants.

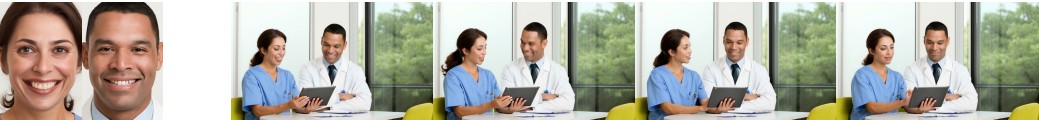

The video depicts two individuals seated at a table, engaged in a discussion while looking at a tablet device. The person on the left is dressed in blue scrubs, indicative of a healthcare professional, and is holding the tablet. The individual on the right is wearing a white lab coat over a collared shirt and tie, suggesting they are a doctor or medical professional.

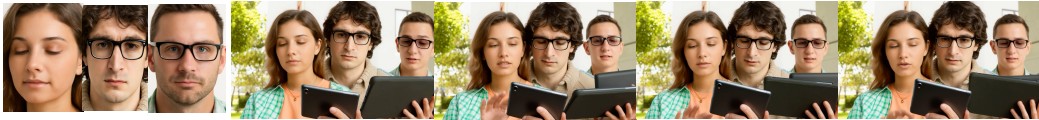

Three individuals seated outdoors on a bench, engaged with electronic devices. The person on the left, wearing a green and white checkered shirt, is focused on a smartphone. The individual in the middle, dressed in a beige sweater, holds a tablet and looks intently at it. The person on the right, clad in a peach-colored cardigan and sunglasses hanging around her neck…

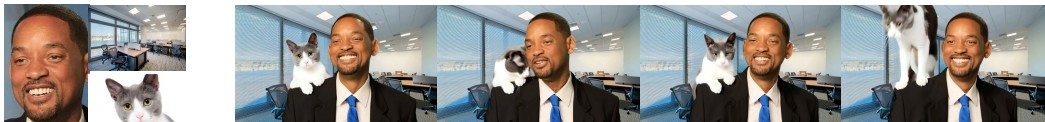

A man sitting in the office, a cat sitting on his legs.

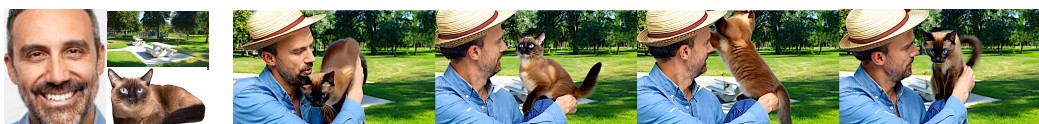

A man sitting in the park, a cat walking around his feet.

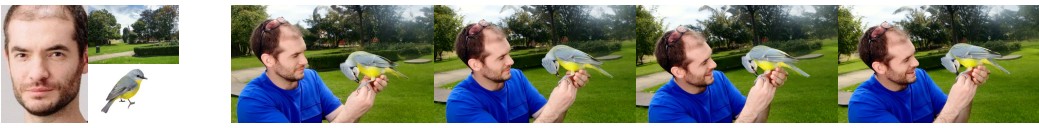

A man feeding a bird in the park.

Figure 14: **Qualitative results of our method on test cases involving multiple concepts.** Such as persons, animals, and scenes. Our model is capable of understanding and encoding multiple subjects based on the reference images.

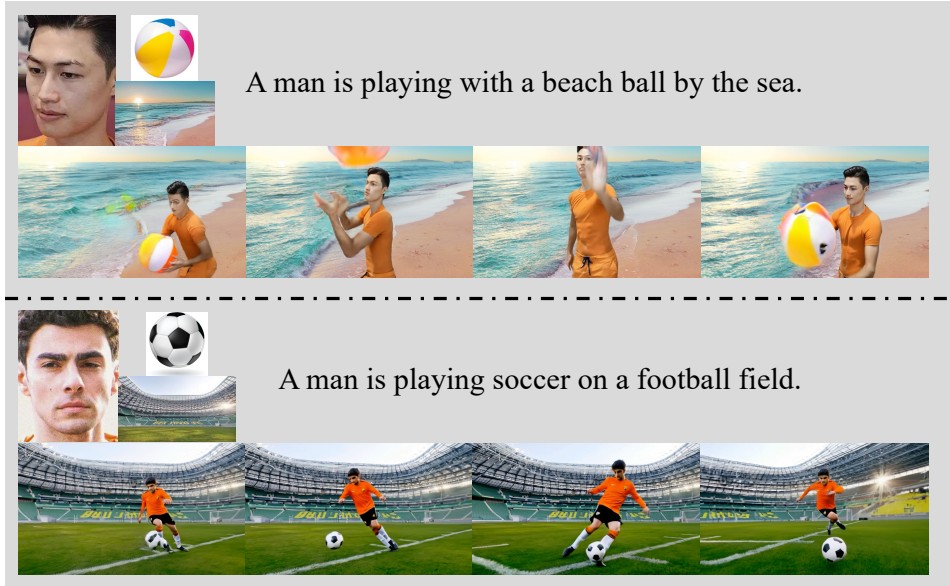

Figure 15: Failure cases.

may produce structural breakdowns, such as incorrect object manipulation or unnatural physical behavior. These issues are not unique to our approach but are commonly observed across existing methods. Addressing these challenges will require the development of richer datasets that capture more complex subject interactions, along with more advanced foundation models that can better simulate physical behaviors and dynamics.

## F ADDITIONAL STATEMENT

### F.1 THE USE OF LARGE LANGUAGE MODEL

We leveraged large language models (LLMs), including ChatGPT, to assist with manuscript preparation. Their use was limited to language-related tasks such as grammar and spelling correction, stylistic polishing, and word choice refinement to improve clarity and readability. We also used LLMs to standardize terminology across sections, check the consistency of figure and table captions with the main text, and streamline reference wording and section cross-references. Please note that all scientific ideas, analyses, and conclusions were conceived, verified, and interpreted solely by the authors.

