# OpenReview forum: "MAGREF: Masked Guidance for Any-Reference Video Generation with Subject Disentanglement"
_ICLR.cc/2026/Conference — ICLR 2026 Poster_

### Official Review · Reviewer_pDBr · 2025-10-30

**Soundness:** 4
**Presentation:** 4
**Contribution:** 3
**Rating:** 6
**Confidence:** 5

**Summary:**

This paper proposes MAGREF, a unified framework for any-reference video generation, addressing challenges such as identity inconsistency, subject entanglement, and copy-paste artifacts. The method integrates masked guidance and subject disentanglement mechanisms, along with a four-stage data pipeline. Experiments show that MAGREF achieves superior performance over existing methods, demonstrating its effectiveness for scalable for  subject-driven video synthesis.

**Strengths:**

The citations are comprehensive, suggesting that the authors are well-versed in the field of subject-driven generation.


The experiments are extensive, encompassing various open-source and commercial baselines, including both single-subject and multi-subject settings. These results clearly demonstrate the effectiveness of MAGREF.

**Weaknesses:**

If the base image-to-video model does not use channel concatenation to fuse the reference images, the effectiveness of this method would be limited.

**Questions:**

(1)  The overall spatial resolution for all subjects is fixed. During inference, how are reasonable spatial resolutions allocated to different subjects? If a subject is very large, such as the background, this implies that the spatial resolution for other subjects will be scaled down, which may lead to information loss.

(2) In Equation (6), is $M^k_{\text{sub}}$  a typographical error? Should it instead be written as $M^i_{\text{sub}} $?

(3) Is Equation (6) also applied during inference? In addition, can Equation (6) be extended to scenarios involving subject interactions, where the subject masks may overlap?

Addressing my concerns would lead me to reconsider and potentially raise my score.

---

> ### Author Response · Authors · 2025-11-21
> **Response to Reviewer pDBr (1/3)**
>
> We sincerely thank the reviewer for the valuable comments and recognition of our work. We especially appreciate the reviewer's pointing out of the typographical errors. We have provided the following detailed responses to the specific issues.
>
> >### **Weakness 1: Limitations in Base I2V Model**
>
> We appreciate the reviewer’s valuable feedback and take this opportunity to provide clarification.
>
> - **The effectiveness of our method is not limited to base models** that use channel concatenation as a fusion strategy. Its applicability is far broader than the reviewer’s concern. Currently, there are two main approaches for processing the first-frame reference image in I2V models: one is the channel concatenation method, represented by Wan[1], and the other is the first-frame replacement method, represented by HunyuanI2V[2]. Both methods encode the first-frame reference image using a Variational Autoencoder (VAE), our method can be applied to any I2V model that uses first-frame reference image encoding.
> - **Our contribution** lies in cleverly utilizing the first frame of the I2V model as a canvas to store visual entities. By combining **masked guidance** and **subject disentanglement**, we efficiently extend the model to support any-reference video generation without altering the model architecture. This reveals the **overlooked yet powerful potential** of I2V-based models for reference content customization.
> - To validate that our method is not dependent on the channel concatenation approach, we conduct a comparative experiment as shown in **Table A** below. We select PUSA v1.0[3] as a comparison, whose I2V model is derived from the fine-tuning of Wan2.1 T2V, employing a similar first-frame replacement strategy. We use 8 H100 GPUs and train on a subset of data for the same number of steps until convergence. The experimental results show that the final outcomes for both the Wan-based channel concatenation and first-frame replacement strategies are comparable, thus proving the transferability of our method.
>
> - Through these experiments, we further validate the generalizability and effectiveness of our approach, demonstrating that it is not only applicable to models using channel concatenation but also achieves similar results in models employing first-frame replacement strategies.
>
> **Table A. Comparison of Different I2V Concatenation Methods**
> | **Method**           | **ID-Sim** | **Subj-Sim** | **Bg-Sim** | **Aesthetic** | **Motion** | **GmeScore** | **Total Score** |
> |----------------------|------------|--------------|------------|---------------|------------|--------------|-----------------|
> | PUSA-based I2V       | 0.492      | 0.450        | 0.487      | 0.481         | 0.882      | 0.652        | 0.574           |
> | Wan-based I2V        | 0.496      | 0.447        | 0.506      | 0.470         | 0.912      | 0.637        | 0.578           |
>
> **References List**
>
> [1] Wan T, Wang A, Ai B, et al. Wan: Open and advanced large-scale video generative models. *arXiv preprint arXiv:2503.20314*, 2025.
> [2] Kong W, Tian Q, Zhang Z, et al. Hunyuanvideo: A systematic framework for large video generative models. *arXiv preprint arXiv:2412.03603*, 2024.
> [3] Liu Y, Ren Y, Artola A, et al. Pusa v1.0: Surpassing wan-i2v with $500 training cost by vectorized timestep adaptation. *arXiv preprint arXiv:2507.16116*, 2025.

---

> ### Author Response · Authors · 2025-11-21
> **Response to Reviewer pDBr (2/3)**
>
> >### **Question 1: Clarification on Spatial Resolution Allocation During Inference**
>
> We thank the reviewer for raising this insightful question. We fully agree with the concerns highlighted, and below is our detailed response to this issue.
>
> Firstly, our design philosophy always adheres to the principle of **"simple but effective"**. Therefore, when handling different entities, we do not design a specific spatial resolution for each entity. Instead, we adopt a more general strategy, where the canvas is evenly divided based on the number of reference images, allowing the model to learn a reasonable spatial layout during end-to-end training. To avoid overly rigid constraints, we randomly shuffle the positions of reference images during training. We do not aim to predefine the relative positions or sizes of each entity on the input canvas, as such an approach would be overly rule-based and limit the model's ability to adapt. Ultimately, both qualitative and quantitative experimental results demonstrate that our method can effectively adjust the spatial layout and size of the reference images, generating high-quality videos.
>
> - **On Spatial Allocation Strategy:** In the current implementation, we have adopted a simple yet effective strategy by allocating roughly equal space to each entity (including the background) on the canvas. While this approach may seem straightforward, it ensures that when the number of entities increases, no entity loses important information due to inadequate space. However, we acknowledge that a more intelligent spatial allocation strategy, such as one based on the importance of textual descriptions or the saliency of images, is undoubtedly an avenue for future exploration.
>
> - **Information Loss and Model Robustness:** The issue of "loss of visual details due to reduced entity resolution" that you raised does not occur in our current setup. Our model is built upon the i2v baseline, where each entity's spatial resolution is inherently small, yet its generation ability remains effective in maintaining visual information from the first frame. Currently, even though the number of reference entities has been expanded to 8-10, the space allocated to each entity is comparable to the ratio in i2v, and thus no significant resolution reduction issue arises.
>
> To more rigorously validate our method's robustness under this challenge, we conduct a scalability test on the reference entities. The **"Ref Scale"** refers to the proportion of the reference image relative to the canvas after division. The experimental results (as shown in **Table B**) indicate that the size of the reference entity resolution does not significantly affect the final generated result. Even when the spatial resolution of a single entity is reduced to 1/8 of the entire canvas, the latent variables encoded by the VAE still retain sufficient identity information, ensuring the entity's recognition and visual consistency. This result demonstrates the effectiveness of our method in handling changes in spatial resolution. Relevant qualitative cases have been provided in the **Appendix Figure 7** for the reviewer's reference.
>
> We believe that these explanations sufficiently address the reviewer's concerns and demonstrate the robustness of our method in handling spatial resolution allocation and information retention.
>
> **Table B. Reference image scalability experiment**
> | **Ref Scale** | **ID-Sim** | **Aesthetic** | **Motion** | **GmeScore** | **Total Score** |
> |---------------|------------|---------------|------------|--------------|-----------------|
> | 1             | 0.529      | 0.510         | 0.864      | 0.708        | 0.653           |
> | 1/2           | 0.559      | 0.511         | 0.942      | 0.691        | 0.676           |
> | 1/3           | 0.528      | 0.493         | 0.927      | 0.701        | 0.662           |
> | 1/4           | 0.547      | 0.504         | 0.936      | 0.706        | 0.673           |
> | 1/6           | 0.554      | 0.482         | 0.941      | 0.697        | 0.669           |
> | 1/8           | 0.543      | 0.492         | 0.937      | 0.696        | 0.667           |

---

> ### Author Response · Authors · 2025-11-21
> **Response to Reviewer pDBr (3/3)**
>
> >### **Question 2 and 3: Clarifications on Typographical Error and Application of Equation (6)**
>
> Thank you for the reviewer’s careful reading and valuable comments. We provide the clarifications as follows.
>
> - **Equation (6) contains a typographical error.** We appreciate the reviewer for pointing this out. The correct notation should be $ M_{\text{sub}}^{i} $, indicating that each value embedding is applied to its corresponding subject mask. This has been corrected in our revised manuscript.
>
> - **Equation (6) is consistently applied during inference.** The same operation is used at inference time: subject-related text tokens are linked to the spatial regions of the composite reference frame, ensuring coherent alignment between textual semantics and visual identity cues throughout the generation process.
>
> - **Only the first frame canvas requires semantic–visual binding.** This binding is performed solely on the first frame canvas, as subsequent frames are generated by the video diffusion model without known future spatial positions. Since all frames share a unified latent representation space, applying the binding on the first frame is sufficient to guide subject–text attention across the entire video.
>
> - **Subject masks do not overlap in our composite reference canvas construction.** The composite reference canvas is manually assembled using non-overlapping regions for different subjects, so mask interference does not occur. As a result, Equation (6) does not require additional handling for overlapping cases. The final qualitative cases presented also demonstrate this, as they generate reasonable multi-subject interaction videos.
>
> We thank the reviewer again for these precise and helpful questions.

---

### Official Review · Reviewer_B9UU · 2025-10-31

**Soundness:** 3
**Presentation:** 3
**Contribution:** 3
**Rating:** 8
**Confidence:** 4

**Summary:**

This paper proposes MARGREF,  a unified masked guidance design by combining region-aware masking with pixel-wise channel concatenation, to inject reference cues at the channel level. It further proposes a subject disentanglement mechanism that maps text semantics to their corresponding visual regions, cleanly separating identities and mitigating cross-reference confusion without extra identity modules. The model achieved SOTA performance in consistent subject-driven video generation.

**Strengths:**

- This paper proposed a novel structure to condition video generation on multiple images by combining multiple images into one. It also proposed data-pipeline to collect large-scale
- The proposed subject disentanglement mechanism is novel and effective in text-prompt alignment for different subjects in the image.
- The results show that the proposed framework is better than other baselines. Ablation study proves that all the proposed module is meaningful.
- The paper is well-written and easy to follow.

**Weaknesses:**

The paper is in general good, some minor points:
- The computational power is not mentioned e.g. how many GPUs have been used
- No details about the dataset e.g. source/size and possible privacy problem for human face data

**Questions:**

- What is the value of C_m?
- What is the base T2V model that MAGREF is fine-tuned on?
- For the composed reference image, does the resolution matter?
- For the composed reference image, does the organization matter?
- In 4.3, what is result in Table 3 / 4 tested on? the score seems different from Table 1 / 2?

**Details Of Ethics Concerns:**

hidden concern for use of human videos

---

> ### Author Response · Authors · 2025-11-21
> **Response to Reviewer B9UU (1/2)**
>
> We sincerely thank the reviewer for your insightful questions and thoughtful ideas about our method. We hope that our point-by-point responses address your concerns.
>
> >### **Weakness 1 and Question 5: Lack of Computational Power Details and Discrepancy in Experimental Results**
>
> Thank you for your questions regarding the details and the tables.
>
> We mention in line 405 of the main text that “All experiments are conducted using NVIDIA H100 80GB GPUs.” Below, we provide further clarifications.
>
> - For the main experiments, we used 64 NVIDIA H100 GPUs for training. **Table 1** and **Table 2** present the results of our main experiments under the single-ID and multi-subject configurations, respectively.
>
> - **Table 4** shows the ablation study for the entire pipeline, which follows the same configuration as the main experiments, i.e., using the full dataset and 64 H100 GPUs for experimentation.
>
> - **Table 3** presents the ablation study for the training paradigm and masking strategies. As stated in Line 446 of the paper, “We validate all methods on a small-scale dataset with equal training steps and use the same training resources to ensure fairness.” To quickly validate the effectiveness and ensure a fair comparison, we used 16 H100 GPUs for experimentation on one subset of the data, which is why the metrics differ slightly from those of the full-scale training in the main experiments.
>
> >### **Weakness 2: Lack of Dataset Details and Potential Privacy Concerns**
>
> Thank you for your concern regarding the dataset. We would like to provide further clarification.
>
> We confirm that the video materials in the training dataset we used (e.g., NexData) have legal copyrights. The dataset is legally purchased from a third-party provider and comes with a commercial use license. The faces involved have been authorized for use in AI training. Therefore, the use of these video data does not involve any privacy issues. Additionally, the dataset’s licensing agreement explicitly allows it to be used for AI training and related research, in compliance with relevant laws and regulations. We will include this information in the revised manuscript.
>
> >### **Question 1 and 2: Clarifications on C_m Value and Base Model**
>
> Thank you for these detailed questions. They are important for improving the clarity and completeness of our paper. We provide the consolidated responses as follows.
>
> - **Value of C_m.** The value of C_m is 4, which follows the default configuration of the Wan2.1-I2V 14B backbone on which our method is built.
>
> - **Base T2V/I2V model used for fine-tuning.** MAGREF is fine-tuned on **Wan2.1-I2V 14B**, and we will make this explicit in the revised manuscript.
>
> We appreciate the reviewer’s questions and hope these clarifications strengthen the readability and completeness of the paper.

---

> ### Author Response · Authors · 2025-11-21
> **Response to Reviewer B9UU (2/2)**
>
> >### **Question 3 and 4: Clarifications on Resolution and Organization**
>
> We thank the reviewer for the valuable question. We would like to take this opportunity to elaborate on the matter.
>
> - **Resolution does not affect the quality of video generation.**
>   - For different subjects, we do not specifically design their corresponding spatial resolution. Instead, we adopt a more general approach, where the canvas is evenly divided into n blocks based on the number of reference images, and each reference subject is adaptively resized to cover the corresponding block. Our goal is clear: we aim for the model to learn the appropriate size and layout of the ground truth video in an end-to-end manner.
>   - **Our method is capable of generating reasonable videos at different resolutions**. The model inherits from the I2V baseline model, where the spatial resolution of each subject element is inherently small. Nevertheless, the generation capability still effectively preserves the visual information of the first frame. Even when the number of reference subjects is extended to 8 or 10, the canvas area occupied by a single subject is similar to the ratio seen in I2V models, so no significant resolution reduction issues arise.
>   - To more rigorously verify the robustness of our method under this challenge, we conduct a scalability test on the reference subjects. The "ref scale" refers to the proportion of the reference image relative to the canvas after division. The experimental results (as shown in **Table A**) indicate that the resolution size of the reference subjects does not significantly impact the final generated output. Even when the spatial resolution of a single subject is reduced to 1/8 of the entire canvas, the latent variables encoded by the VAE still retain sufficient identity information, ensuring subject recognition and visual consistency. This result demonstrates the effectiveness of our method when handling changes in spatial resolution. Relevant qualitative examples are provided in the **Appendix Figure 7** for the reviewer’s reference.
>
> - **Organization does not affect the relative position of subjects in the generated video**
>   - The layout of subjects in the composed reference image does not determine the relative positions in the generated video, which is an intentional design choice. **Our goal is for the model to learn a reasonable spatial layout end-to-end**, rather than relying on fixed input arrangements. To achieve this, we randomly shuffled the positions of the subjects in the reference image during training, decoupling identity information from position information.
>
>   - To validate this, we designed an experiment using MAGREF to infer a task with two subjects (person1 and person2). We used 50 random seeds, with person1 fixed on the left side of the reference image and person2 fixed on the right side.
>
>   - As shown in **Table B**, the experiment results show that when no specific relative position is provided, the left-right position relationship in the generated video has about a 50% chance of occurring, independent of the positions in the reference image. However, when the relative positions are explicitly specified via the prompt, the generated positions align closely with the requested positions.
>
> **Table A. Reference image scalability experiment**
> | **Ref Scale** | **ID-Sim** | **Aesthetic** | **Motion** | **GmeScore** | **Total Score** |
> |---------------|------------|---------------|------------|--------------|-----------------|
> | 1             | 0.529      | 0.510         | 0.864      | 0.708        | 0.653           |
> | 1/2           | 0.559      | 0.511         | 0.942      | 0.691        | 0.676           |
> | 1/3           | 0.528      | 0.493         | 0.927      | 0.701        | 0.662           |
> | 1/4           | 0.547      | 0.504         | 0.936      | 0.706        | 0.673           |
> | 1/6           | 0.554      | 0.482         | 0.941      | 0.697        | 0.669           |
> | 1/8           | 0.543      | 0.492         | 0.937      | 0.696        | 0.667           |
>
> **Table B. Effect of explicit position prompts on left-right spatial relationship**
> | **Experiment Condition**      | **Left Position (%)** | **Right Position (%)** |
> |-------------------------------|-----------------------|------------------------|
> | **No specific prompt**         | 23/50                 | 27/50                 |
> | **Prompt: Left specified**     | 46/50                 | 4/50                  |
> | **Prompt: Right specified**    | 2/50                 | 48/50                  |

---

> > ### Comment · Reviewer_B9UU · 2025-11-21
> > **Thanks for the reply**
> >
> > I have no more questions. And I will keep my score and lean towards accepting this paper.

---

> > > ### Author Response · Authors · 2025-11-21
> > > **Response to Reviewer B9UU for Your Feedback**
> > >
> > > Dear reviewer B9UU,
> > >
> > > We would like to express our sincere gratitude to Reviewer B9UU for acknowledging our work and providing constructive suggestions. We are glad to have addressed your concerns, and we appreciate your positive assessment of the paper. Thank you very much for your consideration in leaning towards accepting the paper.
> > >
> > > Thanks again for the time and effort in reviewing our work.

---

### Official Review · Reviewer_fwvD · 2025-10-31

**Soundness:** 2
**Presentation:** 2
**Contribution:** 2
**Rating:** 4
**Confidence:** 4

**Summary:**

To solve the issue of any-reference video generation task, this paper proposes a unified framework, MAGREF. The framework contains region-aware masking mechanism and subject disentanglement mechanism. Experiments show the scalable, controllable, and high-fidelity any-reference video synthesis results.

**Strengths:**

-  The writing is easy to understand, and the painting is well-drawn.
-  The proposed region-aware masking method preserves subject identity without backbone changes.
-  Experimentally, the paper achieves best single-ID and multi-subject score.

**Weaknesses:**

- 1.	In this paper, multi-subjects are introduced into the videos through a blank canvas. Several subjects are directly added to the canvas with their pixels values. Although this function is useful, my major concerns are listed below:
 -      a)	The canvas size is limited. How many subjects can be placed on the canvas without harming the model’s generation ability?
 -      b)	The positions of these subjects are randomly shuffled during training, in my opinion, the locations of different subjects may contain implicit relationship, such as decide the distance of two subjects in the generated videos. However, this is not discussed in the paper and related ablations are not considered.
 -      c)	Similar to the absolute locations, the scale of each subject in the canvas is still missing. Because such model relies on the explicit vision cues to catch the details of the reference subjects, the scale is an important factor that needs to be considered.
- 2. In subsection “Pixel-wise channel concatenation”, the composited image I_{comp} is encoded by VAE encoder and then concatenated with noised video latents along the channel dimension. I cannot find technically something new that differs from existing methods. Existing methods also apply the pixel-wise image/video with VAE encoder and concatenate the latent with noised video.
- 3. The paper claims that the methods support arbitrary subject categories in Lines 098-103, but in their methods, it is unclear how the model support such ability.
- 4. The qualitative comparison in Fig. 5 seems cannot demonstrate the superiority of the proposed methods, such as compared with close-source method Kling1.6 or the open-source method VACE. Besides, why does the multi-subject result of Skyreels method shows show a poor first frame?
- 5. The experimental results lack more deep analysis to explain why the proposed methods outperform the previous methods in qualitative and quantitative results.
-6. The failure cases and more analyses should be discussed.

**Questions:**

See above weaknesses

---

> ### Author Response · Authors · 2025-11-21
> **Response to Reviewer fwvD (1/4)**
>
> We express our sincere gratitude to the reviewer for the insightful comments and the recognition of our work. Following the feedback, we have added experiments and addressed the specific concerns raised by the reviewer as detailed below. We believe these enhancements significantly contribute to the clarity and impact of our research.
>
> >### **Weakness 1 a) and c): Canvas Limitations and Subject Scale Considerations**
>
> Thank you for your valuable question. Below is our response:
>
> - **Regarding canvas limitations and the number of supported subjects.** In **Table A** below, we present **the number of tokens** in the self-attention mechanism. The default resolution of the generated video is 81x480x832. After VAE downsampling and patchify operations, the latent space consists of 21x30x52 = 32,760 tokens. As shown in **Table A**, our method keeps the computational cost constant as the number of reference images increases, while previous methods experience a significant computational burden when scaling to more reference images (e.g., 8 images).
>
> - **Regarding the subject scale experiment.** We conduct both qualitative and quantitative experiments on scalability. The qualitative cases are shown in the **Appendix Figure 7**. The quantitative results are presented in **Table B** below. We find that the training is not very sensitive to the pixel size of reference subject images. **The overall performance does not significantly change with the size of the reference images**. Specifically, we randomly select 50 reference images from the evaluation set for validation and scale the reference images. The **"Ref Scale"** refers to the ratio of the reference image's area within the entire canvas. We observe an interesting phenomenon: even when the reference image is scaled to 1/8 of the canvas ratio, the information is still well-preserved, which means we can support up to eight reference images while maintaining the original computational cost.
>
> **Table A. Computational token comparison as the number of reference images increases**
>
> | Method         | ref_img = 1          | ref_img = 2          | ref_img = 3          | ref_img = 4          | ref_img = 8          |
> |----------------|----------------------|----------------------|----------------------|----------------------|----------------------|
> | **Token-based** | 32760 + 1560×1        | 32760 + 1560×2        | 32760 + 1560×3        | 32760 + 1560×4        | 32760 + 1560×8        |
> | **Ours**        | 32760               | 32760               | 32760               | 32760               | 32760               |
>
> **Table B. Reference image scalability experiment**
> | **Ref Scale** | **ID-Sim** | **Aesthetic** | **Motion** | **GmeScore** | **Total Score** |
> |---------------|------------|---------------|------------|--------------|-----------------|
> | 1             | 0.529      | 0.510         | 0.864      | 0.708        | 0.653           |
> | 1/2           | 0.559      | 0.511         | 0.942      | 0.691        | 0.676           |
> | 1/3           | 0.528      | 0.493         | 0.927      | 0.701        | 0.662           |
> | 1/4           | 0.547      | 0.504         | 0.936      | 0.706        | 0.673           |
> | 1/6           | 0.554      | 0.482         | 0.941      | 0.697        | 0.669           |
> | 1/8           | 0.543      | 0.492         | 0.937      | 0.696        | 0.667           |

---

> ### Author Response · Authors · 2025-11-21
> **Response to Reviewer fwvD (2/4)**
>
> >### **Weakness 1 b): Implicit Relationships on Subject Positions**
>
> We would like to thank you for your valuable question. Below is our response:
>
> - Our design philosophy adheres to the principle of "**simple but effective**". To avoid overly rigid constraints, we randomly shuffle the positions of reference images during the training phase, allowing the model to learn reasonable spatial layouts through end-to-end training. We do not wish to predefine the relative positions or sizes of each subject in the input canvas, as this approach is too rule-based and limits the model's adaptability. Ultimately, both qualitative and quantitative experimental results show that our method can effectively adjust the spatial layout and size of each reference image, generating high-quality videos.
>
> - To validate this, we designed an experiment where MAGREF infers a task involving two subjects (person1 and person2). Using 50 random seeds, we fixed person1 on the left side of the reference image and person2 on the right side.
>
> - The results in **Table C** show that, when no specific relative position is specified, the left-right relationship in the generated video occurs with a probability of approximately 50%, independent of the position relationship in the reference image. However, when positions are explicitly specified through prompts, the generated relative positions closely align with the prompt's requirements.
>
> **Table C. Effect of explicit position prompts on left-right spatial relationship**
>   | **Experiment Condition**      | **Left Position (%)** | **Right Position (%)** |
>   |-------------------------------|-----------------------|------------------------|
>   | **No specific prompt**         | 23/50                 | 27/50                  |
>   | **Prompt: Left specified**     | 46/50                 | 4/50                  |
>   | **Prompt: Right specified**    | 2/50                 | 48/50                  |
>
> >### **Weakness 2: Difference with Existing Methods**
>
> Thank you for your question, which helps improve the clarity of our paper.
>
> **We would like to clarify that our goal is not to modify the commonly used mechanism of concatenating VAE-encoded latent representations with noised video latents, this approach is indeed standard in existing I2V frameworks. Instead, our innovation lies in enabling this architecture to support any-reference video generation without architectural changes through our proposed *Masked Guidance* and *Subject Disentanglement* algorithms.** We have conducted extensive experiments to illustrate this distinction, as shown in **Table 3** of the main paper and **Figure 6** in the Appendix. Below, we provide a detailed explanation.
>
> - **Limitation of vanilla concatenation in multi-reference settings.** As shown in **Table D** below, directly applying the standard I2V concatenation to any-reference video generation introduces artifacts and degrades the information of the first frame, consistent with the qualitative evidence in **Figure 6**. In contrast, our method preserves the inherent strengths of the I2V backbone and effectively adapts it to any-reference video generation.
>
> - **Challenges of identity consistency in multi-subject generation.** Maintaining identity fidelity is a significant challenge in personalized video generation. Prior works such as ConsisID[1] and FantasyID[2] rely on external priors, while Q-Former introduces feature-injection modules. Although pixel-wise concatenation is common in I2V, it has rarely been applied to subject-driven generation. When directly extended to multi-reference scenarios, as seen in SkyReels-A2, it disrupts inter-frame feature distributions and results in noticeable artifacts.
>
> - **Our contribution: Masked Guidance + Subject Disentanglement.** To address these issues, we propose a **simple yet effective** combination of a region-aware masking mechanism and subject disentanglement. Our method efficiently inherits the native capabilities of the I2V backbone **while resolving identity-preservation and temporal-consistency challenges** inherent to multi-subject generation, as validated by both qualitative and quantitative results.
>
> **Table D. Performance comparison of vanilla I2V concatenation and our method**
>
> | Method                      | D-Sim | Subj-Sim | Bg-Sim | Aesthetic | Motion | GmeScore | Total Score |
> |-----------------------------|-------|----------|--------|-----------|--------|----------|-------------|
> | Vanilla I2V Concatenation   | 0.458 | 0.431    | 0.492  | 0.437     | 0.876  | 0.653    | 0.558       |
> | Ours                        | 0.504 | 0.452    | 0.526  | 0.452     | 0.906  | 0.679    | 0.587       |
>
>
> [1] Yuan, Shenghai, et al. "Identity-preserving text-to-video generation by frequency decomposition." Proceedings of the Computer Vision and Pattern Recognition Conference. 2025.
>
> [2] Zhang, Yunpeng, et al. "Fantasyid: Face knowledge enhanced id-preserving video generation." arXiv preprint arXiv:2502.13995 (2025).

---

> > ### Author Response · Authors · 2025-11-21
> > **Response to Reviewer fwvD (3/4)**
> >
> > >### **Weakness 3: Lack of Clarity on Supporting Arbitrary Subject Categories**
> >
> > Thank you for pointing this out, we provide our response below.
> >
> > - Our approach decouples the image subject condition and the word condition, allowing us to flexibly match subject categories with condition images during training. This method is seamlessly aligned with the I2V (Image-to-Video) backbone training framework, where the subject information is disentangled from other conditional factors. By leveraging this disentanglement, our model is capable of associating arbitrary subject categories with the corresponding visual features, even when those categories were not explicitly part of the training set.
> >
> > - **This ability extends to zero-shot generation**, meaning the model can generate realistic video content for subjects it has never encountered before, simply by conditioning on a new subject description. For example, by providing a textual description of a new subject category, the model can generate consistent and plausible results, as evidenced by both the teaser image and the qualitative cases presented in the Appendix. These examples highlight the model's generalization ability and robustness in handling unseen subject categories, demonstrating that it can adapt to a wide range of conditions without requiring additional retraining or fine-tuning.
> >
> > Thus, our method not only supports arbitrary subject categories but does so in a manner that is both efficient and scalable, maintaining high-quality output even in zero-shot settings.
> >
> > >### **Weakness 4: Insufficient Demonstration of Superiority and SkyReels' Poor First Frame**
> >
> > Thank you for your detailed observations and questions. We haved included more qualitative cases in the **supplementary materials** to further highlight our results. We would like to provide the following explanations:
> >
> > - **Regarding the qualitative results in Figure 5.** As you pointed out, both Kling1.6 and the open-source method VACE maintain a high visual similarity with the provided reference images. However, upon closer inspection, there are notable issues: in the VACE-generated case, the man’s hair noticeably elongates and becomes distorted, while in the Kling-generated case, the man’s hand deforms in the subsequent frames, which is illogical. In contrast, our method successfully preserves important details, such as facial skin tone, the white ceramic background, and the black gourd.
> >
> > - **Regarding the initial frames in SkyReels.** We present this experiment in detail in **Figure 6 of the Appendix**. As you mentioned in Question 2, SkyReels actually uses the conventional approach of concatenating latent reference images with video noise. When dealing with multiple reference images, this approach can potentially disrupt the feature space of the original I2V backbone. Additionally, insufficient training leads to blurry and artifact-laden frames in the early stages. Our method effectively addresses this issue.

---

> ### Author Response · Authors · 2025-11-21
> **Response to Reviewer fwvD (4/4)**
>
> >### **Weakness 5: Lack of In-depth Analysis of Experimental Results**
>
> Thank you for your valuable question and suggestion. Due to space constraints in the previous version, we placed the detailed analysis in the Appendix. In the revised version, we will further expand the in-depth analysis of the experimental results in the main body. Below is our detailed response:
>
> Our method addresses the existing issues in generating custom videos with any-reference through three major innovations:
> **(1) Pixel-level channel concatenation and region-aware masking mechanism**, ensuring consistent multi-subject identity preservation. **(2) Subject disentanglement mechanism**, reducing cross-subject confusion. **(3) Four-stage data pipeline design**, integrating filtering, annotation, object and face processing, effectively mitigating copy-paste artifacts. Through these innovations, **MAGREF** offers a scalable, controllable, and high-fidelity solution for any-reference video generation.
>
> - **Analysis on masking strategy and concatenation scheme.** We provide a detailed analysis in **Appendix D.1** and the experiment in **Figure 6**. As described, the region-aware masking mechanism explicitly adjusts the spatial and temporal contribution of each reference image. By masking irrelevant or redundant areas and retaining only task-relevant cues, the model avoids channel-level entanglement and significantly reduces cross-subject interference. This enables the generator to better utilize fine-grained visual information while maintaining consistency with the I2V training paradigm.
>
> - **Analysis on subject disentanglement mechanism.** We show the interpretability of the subject disentanglement mechanism in **Figure 3** and the ablation experiment in **Table 4**, demonstrating that explicitly binding each subject to its corresponding textual condition effectively reduces interference and improves multi-subject video generation quality. Further explanations are also provided in **Appendix D.2**.
>
> - **Analysis on data processing pipeline.** In **Table 4**, we conducted an ablation experiment on the data pipeline, showing that removing cross-pair data significantly degrades overall performance, especially in reducing copy-paste artifacts. This confirms that the cross-pair augmentation strategy helps mitigate such artifacts by enriching subject–text associations.
>
>
> >### **Weakness 6: Lack of Discussion on Failure Cases and Further Analysis**
>
> Thank you for your attention to the failure cases. Due to page limitations in the main paper, we initially placed this section in the Appendix. As shown in **Figure 14** and **Appendix E.2**, our model has significant limitations when dealing with physical laws and strong interactions. Of course, this largely depends on the performance of the underlying model.
>
> We appreciate your reminder, and we will include a detailed analysis of the failure cases in the revised version of the main paper.

---

### Official Review · Reviewer_YBDZ · 2025-11-01

**Soundness:** 3
**Presentation:** 3
**Contribution:** 3
**Rating:** 6
**Confidence:** 5

**Summary:**

The paper introduces MAGREF, a framework for any-reference video generation that combines region-aware masked guidance with subject disentanglement to support arbitrary combinations of human, object, and environment references. It achieves superior identity preservation and visual quality compared to open-source and proprietary baselines.

**Strengths:**

1. Technically sound: Masked guidance and pixel-wise concatenation are simple yet effective extensions of I2V backbones.

2. Comprehensive results: The experiments and ablations clearly support the claimed improvements.

**Weaknesses:**

1. My major concern is that the pixel-wise channel concatenation may limit scalability when the number of reference subjects grows. It is unclear how the model handles more subjects simultaneously. Would temporal or latent-level concatenation yield more flexible conditioning in such cases?
2. While pixel-wise concatenation effectively preserves subject appearance, it may inherently limit global-level customization. Since it injects spatially grounded features, the model mainly captures concrete subject geometry rather than abstract global styles (e.g., tone, lighting, art style). Temporal or latent-level fusion could offer more flexibility for such style control. Can this framework be extended to global element customization, such as atmosphere, or texture?

**Questions:**

As seen in weakness

---

> ### Author Response · Authors · 2025-11-21
> **Response to Reviewer YBDZ (1/2)**
>
> We sincerely thank the reviewer for the insightful comments and recognition of this work, we have clarified the below points and added the needed quantitative and qualitative experiments.
>
> >### **Weakness 1: Scalability of Reference Subjects**
>
> We first thank the reviewer for this valuable concern. We clarify that the design we have adopted does not introduce scalability bottlenecks; on the contrary, it offers greater flexibility while maintaining computational efficiency.
>
> - **Computational efficiency.** Most existing methods support a maximum of 3 or 4 reference images. A key reason for this limitation is that existing methods based on temporal or latent-level concatenation face significant computational burdens when scaling to more reference images (e.g., 8 images). As shown in **Table A** below, we define this as token-based methods. We compute **the number of tokens** performing full attention in DIT under the default configuration, where the generated video’s resolution is 81x480x832. After VAE downsampling and patchification, the latent space contains 21x30x52 = 32,760 tokens. As the number of reference images increases, the number of tokens rapidly grows, and the computational load increases logarithmically with log(n*2), surpassing the original computational cost once a certain threshold is reached. **In contrast**, our method is based on a complete canvas, and with the addition of reference images, the computational load remains constant, demonstrating strong computational efficiency.
>
> - **Scalability with reference images.** Building on this, we conduct both qualitative and quantitative experiments to assess scalability. The qualitative cases are shown in the **Appendix Figure 7**. Quantitatively, as shown in **Table B** below, we randomly select 50 reference images from the evaluation set for validation and scale them. The "ref scale" refers to the proportion of the reference image relative to the canvas after division. We observe an interesting phenomenon: even when the reference image is scaled down to 1/8 of the canvas, the information is still well-preserved. This indicates that we can support up to 8 reference images. The overall effect does not show significant changes with the size of the reference images.
> We believe the reason for maintaining this performance is, on the one hand, the I2V training paradigm effectively preserves identity features, and on the other hand, even when scaled to 1/8, it likely corresponds to the approximate proportion the subject occupies in real-world videos, which minimizes information loss.
>
> **Table A. Computational token comparison as the number of reference images increases**
>
> | Method         | ref_img = 1          | ref_img = 2          | ref_img = 3          | ref_img = 4          | ref_img = 8          |
> |----------------|----------------------|----------------------|----------------------|----------------------|----------------------|
> | **Token-based** | 32760 + 1560×1        | 32760 + 1560×2        | 32760 + 1560×3        | 32760 + 1560×4        | 32760 + 1560×8        |
> | **Ours**        | 32760               | 32760               | 32760               | 32760               | 32760               |
>
> **Table B. Reference image scalability experiment**
> | **Ref Scale** | **ID-Sim** | **Aesthetic** | **Motion** | **GmeScore** | **Total Score** |
> |---------------|------------|---------------|------------|--------------|-----------------|
> | 1             | 0.529      | 0.510         | 0.864      | 0.708        | 0.653           |
> | 1/2           | 0.559      | 0.511         | 0.942      | 0.691        | 0.676           |
> | 1/3           | 0.528      | 0.493         | 0.927      | 0.701        | 0.662           |
> | 1/4           | 0.547      | 0.504         | 0.936      | 0.706        | 0.673           |
> | 1/6           | 0.554      | 0.482         | 0.941      | 0.697        | 0.669           |
> | 1/8           | 0.543      | 0.492         | 0.937      | 0.696        | 0.667           |

---

> > ### Author Response · Authors · 2025-11-21
> > **Response to Reviewer YBDZ (2/2)**
> >
> > >### **Weakness 2: Limitation for Global Customization**
> >
> > First, thank you for your insightful observations. We would like to provide our response here.
> >
> > - The point you raised about providing more flexibility for abstract global style control is very interesting. We have also recently started exploring this interesting combination. While some recent works in image generation[1] have combined such abstract global style control, there has been no similar progress in the video generation domain yet.
> >
> > - Since our current work mainly focuses on comparisons with prior methods regarding entity subjects, we consider incorporating global style control as part of our future work. In our view, **the main challenge lies in constructing the corresponding paired video data**, and we are actively working on creating such datasets.
> >
> > - At the methodological level, **our approach can be naturally extended to decouple style from subjects**. Specifically, the Subject Disentanglement module can be further extended into a Style Disentanglement module, by injecting the attention values of style-related words from reference images to enable arbitrary global element customization. Then, a progressive blending training strategy can be adopted, where the subject and style are first learned separately, and then jointly modeled.
> >
> > Once again, we sincerely thank the reviewer for the insightful comment and look forward to further discussions.
> >
> > [1] Garibi D, Yadin S, Paiss R, et al. Tokenverse: Versatile multi-concept personalization in token modulation space[J]. ACM Transactions On Graphics (TOG), 2025, 44(4): 1-11.

---

> > > ### Author Response · Authors · 2025-11-26
> > > **Further Response to Reviewer YBDZ Regarding Global Customization (2/2 Supp)**
> > >
> > > >### **More Promising Experiments for Exploring the Global Customization Capability**
> > >
> > > Once again, thank you for your insightful comments on the global customization capability of our method.
> > > - Following your suggestion, **we have conducted new qualitative experiments** to explore the generalization ability of our method under abstract global styles, such as tone and lighting. The results are promising. As shown in **Appendix Figure 8**,  the revised content has been highlighted in blue, our method can generate high-quality videos with lighting or atmospheric reference images in a zero-shot manner, even without having seen abstract style training data.
> > > - We believe this ability arises from our proposed **Subject Disentanglement** mechanism, which cleverly injects subject tokens into reference images, further emphasizing the relevant elements needed from the reference. While the results for generating such abstract styles are still somewhat unstable, it demonstrates the potential of our approach and motivates us to further explore a more general, true "any-reference" video generation.
> > >
> > > Once again, thank you for your valuable and meaningful suggestions. We look forward to your feedback and further discussions!

---

### Author Response · Authors · 2025-11-21
**General Response to All Reviewers**

We sincerely thank all the reviewers for their constructive comments and recognition of this work. We have taken all suggestions seriously and conducted extensive experiments. The revised content has been highlighted in blue in the updated manuscript. Below is a summary of the main responses and improvements:

1. Added a quantitative table on reference image scalability experiment.
2. Included a qualitative experiment on reference image scalability in Appendix Figure 7.
3. Analyzed a computational token comparison as the number of reference images increases.
4. Added a table comparing different I2V concatenation methods.
5. Added a table showing the effect of explicit position prompts on the left-right spatial relationship.
6. Added more qualitative comparison cases in the supplementary materials.

We hope our responses below convincingly address all reviewers’ concerns. We thank all reviewers’ time and efforts again!

---

### Meta-Review · Area_Chair_aw1L · 2026-01-07

**Summary:**

This paper tackles  the task of any-reference video generation. Below are reviewers' concerns.
1. Reviewer YBDZ raises concerns about pixel-wise channel concatenation, which may limit scalability an inherently limit global-level customization.
2. Reviewer fwvD raises concerns that the canvas is limited, the locations of different subjects may contain implicit relationship, and lack of sufficient analysis of experiment results.
3. Reviewer B9UU is in general satisfied with this paper. His minor concerns are the lack of details on computation resource and dataset.
4. Reviewer pDBr's major concern is that the proposed method would be limited if the base video model does not use channel concatenation to fuse reference images.

Overall, most reviewers acknowledge that the results show superiority of this paper against existing work, and the proposed method is technically solid.

**Reviewer Concerns:**

1. The rebuttal conducted additional experiments on Reference Image Scalability and global custoization ability of their model, which I believe has addressed reviewer YBDZ's major concerns.
2. Reviewer fwvD's concerns about the limited canvas is not fully addressed in the rebuttal.
3. The rebuttal conducted  quantitative experiments on different I2V concatenation methods, demonstrating the flexibility of their model. This addresses reviewer pDBr's major concern.

**Reviewer Scores:**

The scores seem not to have been changed after the rebuttal.

---

### Decision · Program_Chairs · 2026-01-26

Accept (Poster)